# Transmission of MPXV from fire-footed rope squirrels to sooty mangabeys

Carme Riutord-Fe[1,2,15], Jasmin Schlotterbeck[1,15], Lorenzo Lagostina[1], Leonce Kouadio[2,3], Harriet R. Herridge[1], Moritz J. S. Jochum[1], Nea Yves Noma[1,4], Ane López-Morales[2], Donata Hoffmann[5], Sten Calvelage[5], Hjalmar Kühl[4,6], Alexander Mielke[7], Catherine Crockford[2,8], Liran Samuni[2,9], Roman M. Wittig[2,8], Martin Beer[5], Sery Gonedelé-Bi[10], Jan F. Gogarten[11,12], Sébastien Calvignac-Spencer[12,13,14,15], Ariane Düx[1,2,15], Livia V. Patrono[1,15 ✉] & Fabian H. Leendertz[1,2,12,14,15 ✉]

Mpox, caused by the monkeypox virus (MPXV; *Orthopoxvirus monkeypox*), is on the rise in West and Central Africa[1–3]. African rodents, especially squirrels, are suspected to be involved in MPXV emergence, but no evidence of a direct transmission to humans or non-human primates has been established[4–9]. Here we describe an outbreak of MPXV in a group of wild sooty mangabeys (*Cercocebus atys*) in Taï National Park (Côte d'Ivoire). The outbreak affected one-third of the group, killing four infants. To track its origin, we analysed rodents and wildlife carcasses from the region. We identified a MPXV-infected fire-footed rope squirrel (*Funisciurus pyrropus*), found dead 3 km from the mangabey territory 12 weeks before the outbreak. MPXV genomes from the squirrel and the mangabey were nearly identical. A video record from 2014 showed a mangabey from this group eating the same squirrel species and diet metabarcoding of faecal samples collected from mangabeys before the outbreak identified two samples containing fire-footed rope squirrel DNA. One of these samples was also the first positive for MPXV. This represents a rare case of direct detection of interspecies transmission. Our findings indicate that rope squirrels were the source of the MPXV outbreak in mangabeys. Because squirrels and non-human primates are hunted, traded and consumed by humans in West and Central Africa[10,11], exposure to these animals probably represents risk for zoonotic transmission of MPXV.

The recent emergence of MPXV lineages characterized by human-to-human transmission through sexual networks has led the World Health Organization (WHO) to declare public health emergencies of international concern in 2022 (ref. 1) and 2024 (ref. 2). Efforts were immediately scaled up in endemic African countries to reinforce mpox surveillance systems. Concurrently, sustained human-to-human transmission was shown to leave a distinct signature in MPXV genomes identifiable as APOBEC3-induced mutations, providing a tool to determine how much MPXV evolution happened in humans[12]. Building on these advances, recent genomic surveillance data from the Democratic Republic of Congo (DRC), the Republic of Congo, Nigeria and Cameroon clearly showed that MPXV diversity mostly reflects numerous, independent zoonotic spillovers[13–15]. Importantly, epidemiological data from the DRC suggest that these spillovers increased in frequency from 2010 to 2024, a period during which the national surveillance system has been relatively stable[3].

Identifying the animal(s) that serve as reservoir(s) for MPXV may help in managing the risk of spillover to humans and preventing subsequent outbreaks fuelled by human-to-human transmission. Following others[16], we define a reservoir as a natural host in which the virus can circulate permanently and from which transmission to another host—humans in the case of zoonoses—is possible and documented. According to this definition, no MPXV reservoir has yet been identified. Nevertheless, extensive information has accumulated over the five decades since the virus discovery, pointing to several potentially involved species.

MPXV was first isolated from wildlife in 1985, when a Thomas's rope squirrel (*Funisciurus anerythrus*) captured in the DRC tested positive[7].

[1]Department of Ecology and Emergence of Zoonotic Diseases, Helmholtz Institute for One Health (HIOH), Helmholtz Centre for Infection Research (HZI), Greifswald, Germany. [2]Taï Chimpanzee Project, Centre Suisse de Recherches Scientifiques en Côte d'Ivoire, Abidjan, Côte d'Ivoire. [3]Université Peleforo Gon Coulibaly Korhogo, Korhogo, Côte d'Ivoire. [4]Senckenberg Museum for Natural History Görlitz, Senckenberg—Member of the Leibniz Association, Görlitz, Germany. [5]Institute of Diagnostic Virology, Friedrich Loeffler Institute, Riems, Germany. [6]International Institute Zittau, Technische Universität Dresden, Zittau, Germany. [7]Centre for Brain and Behaviour, Department of Psychology, Queen Mary University of London, London, UK. [8]Ape Social Mind Lab, Institute of Cognitive Sciences, CNRS University of Lyon, Lyon, France. [9]Cooperative Evolution Lab, German Primate Center, Göttingen, Germany. [10]Laboratoire de Biotechnologie, Agriculture et Valorisation des Ressources Biologiques, UFR Biosciences, Université Félix Houphouët-Boigny d'Abidjan-Cocody, Abidjan, Côte d'Ivoire. [11]Evolutionary Community Ecology Research Group, Helmholtz Institute for One Health (HIOH), Helmholtz Centre for Infection Research (HZI), Greifswald, Germany. [12]University of Greifswald, Greifswald, Germany. [13]Department of Pathogen Evolution, Helmholtz Institute for One Health (HIOH), Helmholtz Centre for Infection Research (HZI), Greifswald, Germany. [14]University Medicine Greifswald, Greifswald, Germany. [15]These authors contributed equally: Carme Riutord-Fe, Jasmin Schlotterbeck, Sébastien Calvignac-Spencer, Ariane Düx, Livia V. Patrono, Fabian H. Leendertz. ✉e-mail: liviavictoria.patrono@helmholtz-hioh.de; fabian.leendertz@helmholtz-hioh.de

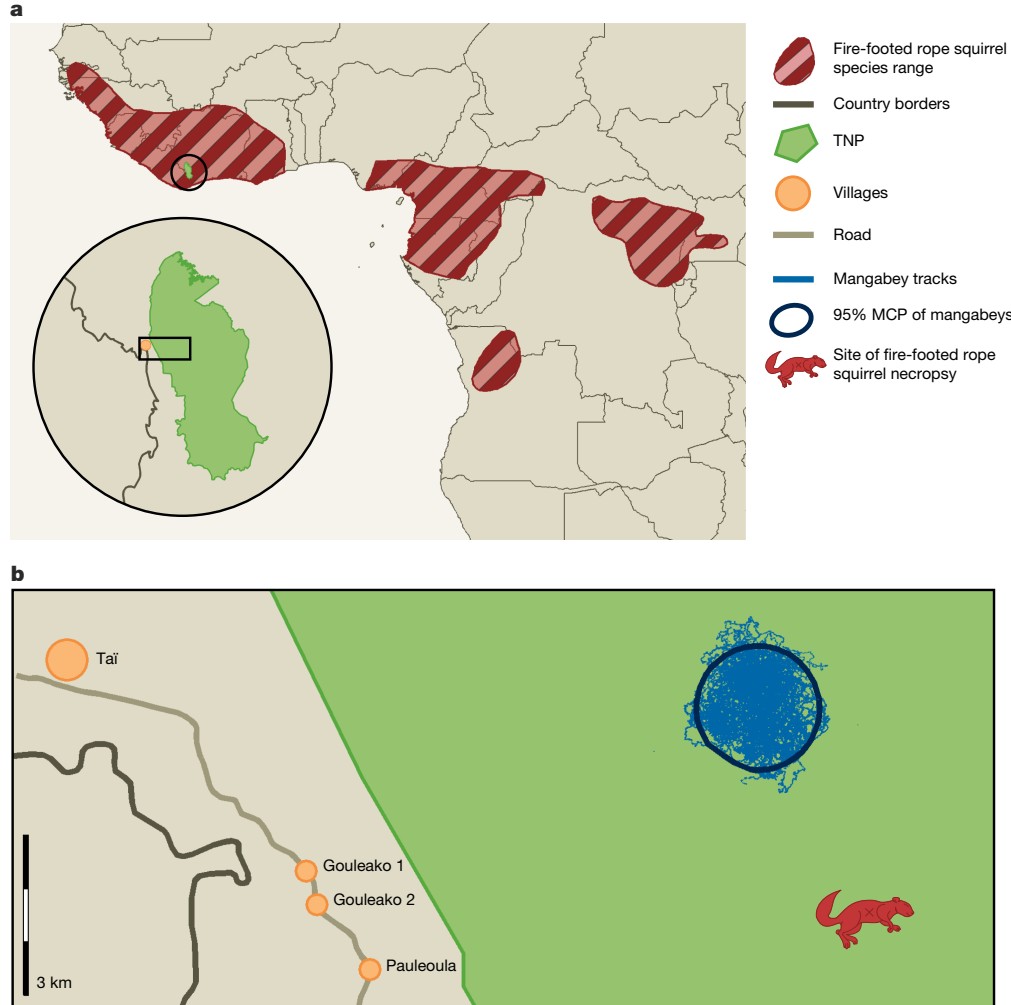

**Fig. 1 | Fire-footed rope squirrel range and map of the sooty mangabey territory in TNP. a**, Distribution of the fire-footed rope squirrel in Africa (striped red) according to IUCN redlist[32] and the location of TNP in Côte d'Ivoire for reference (green). **b**, Home range of the habituated sooty mangabey group at TCP, approximated as the 95% minimum convex polygon (MCP) (dark blue) line of the mangabey movements tracked over the period of a year (light blue). The squirrel icon represents the location of the necropsy of the fire-footed rope squirrel which was found positive for MPXV. Fire-footed rope squirrels are territorial, with typical home ranges of a few hectares. Range sizes[33]: male *F. pyrropus* 5.2 ha, subadult female 2.3 ha, lactating female 1 ha. This squirrel home range would not have overlapped with that of the sooty mangabey study group. Map in **a** adapted from OpenStreetMap (https://www.openstreetmap.org), reproduced under a Creative Commons CC BY-SA 2.0 licence. The dead squirrel icon in **b** was created using Inkscape.

This finding supported the idea that this and other African rodents are natural hosts, a notion previously supported by sero-epidemiological studies indicating orthopoxvirus (OPV) circulation in this group of mammals. PCR analyses of museum specimens later detected MPXV DNA in five rope squirrel species, including *F. anerythrus* (45 of 362; 12.4%) and *Funisciurus pyrropus* (8 of 201; 4%)[9]. Most recently, a large screening of small mammals in the DRC reported near-complete genomes of MPXV derived from a Thomas's rope squirrel, another squirrel (*Paraxerus* sp.) and one soft-furred mouse (*Praomys jacksoni*)[8]. Furthermore, spatial overlap analyses between ecological niches of 99 African mammal species and human mpox index cases similarly pointed to squirrels as the most likely source of human infections[5] (Extended Data Fig. 1 shows rope squirrel species distribution). These repeated and independent findings suggest that (rope) squirrels may be among the natural hosts of MPXV. However, direct evidence of transmission from these animals (or any other potential natural host) to humans or other hosts is still lacking.

By contrast, a plausible chain of events involving African rodents was reconstructed during an MPXV outbreak affecting humans outside the African continent, in the USA in 2003 (ref. 4). All human cases were linked to pet prairie dogs (*Cynomys* sp.) sold by a single distributor, who had previously housed them with several African rodent species imported through a single shipment from Ghana. Among these, at least three species (giant pouched rats—*Cricetomys* sp., rope squirrels—*Funisciurus* sp. and dormice—*Graphiurus* sp.) tested positive for MPXV[6]. Subsequent investigations in Ghana found individuals from the same genera either seropositive or PCR-positive for OPV[17]. Notably, this detailed reconstruction identified only the prairie dogs—an incidental host—as the direct source of human infection, under very specific circumstances that provide little insight into zoonotic transmission in endemic areas. Furthermore, it was not possible to determine the relative contribution of each African rodent species to virus transmission, either among themselves or to prairie dogs. Although these investigations strengthened the suspicion that some African rodents may serve as MPXV reservoirs, none could be formally confirmed.

Captive non-human primates (NHP) were associated with the discovery of MPXV[18], which resulted in the misleading naming of the virus. More recently, long-term health monitoring at the Taï Chimpanzee Project[19] (TCP) in Taï National Park (TNP), Côte d'Ivoire, showed that MPXV also affects wild NHP, opening a window into the ecology of this

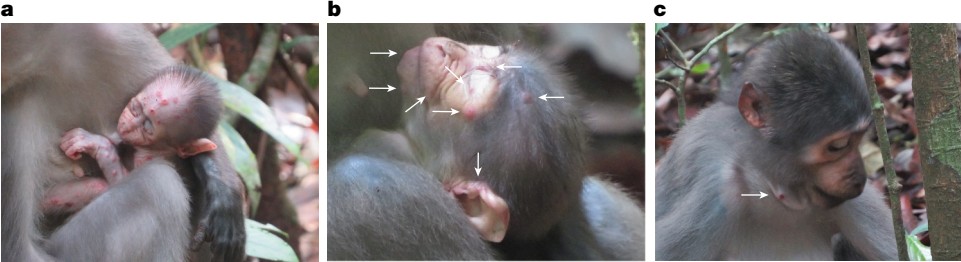

**Fig. 2 | Different degrees of maculopustular rash associated with MPXV infection in three sooty mangabeys. a**, Severe maculopustular rash (more than 20 lesions) spread all over the body. **b**, Moderate maculopustular rash (5–20 lesions) localized on the head. **c**, Mild maculopustular rash (1–5 lesions) localized on the neck. White arrows point at the observed skin lesions. Photo by Taï Chimpanzee Project/Carme Riutord-Fe.

virus in its original sylvatic environment. We detected MPXV in a dead sooty mangabey (hereafter used interchangably with 'mangabey'; *Cercocebus atys*) in 2012 (ref. 20) and later identified three independent MPXV outbreaks in 2017 and 2018 that hit distinct groups of western chimpanzees (*Pan troglodytes verus*) living in the same forest[21]. These studies provided insights into different clinical presentations and described viral genomic diversity in this area, but the source of these outbreaks remains unknown. In late January 2023, we started observing clinical signs compatible with MPXV infection in several infants from the habituated mangabey group, which consisted of 80 individuals at that time (Fig. 1). Sooty mangabeys form stable, cohesive, matrilocal social groups and nearly 30% of the study group developed skin lesions over the following 12 weeks. Making use of the longitudinal non-invasive sample collection started in the early 2010s[22], along with the systematic carcass monitoring[23,24] and rodent sampling efforts in and around TNP, we conducted an outbreak investigation that included an extensive search for a potential source of infection.

On 27 January 2023, an infant mangabey developed red macular lesions on the forehead, back of the head, chest and legs (Fig. 2a), accompanied by the onset of lethargy and anorexia. Lesions quickly spread to the entire body and the individual died within 48 h, on 29 January. By early March, five other infants developed similar lesions alongside lethargy, anorexia and lymphadenopathy. Macular skin lesions progressed to papulopustular stages and three of these infants died. A milder form of the disease, consisting of either a diffuse rash with only about 5–20 skin lesions (Fig. 2b), or fewer isolated lesions appearing in a single part of the body (for example, face, limbs or tail; Fig. 2c), affected 20 other mangabeys of all age groups (Extended Data Table 1). In all affected animals, papulopustular lesions evolved to crusts and ultimately scabs (Extended Data Fig. 2). Overall, the disease swept through the group until the end of April 2023, resulting in 26 out of 80 (32.5%) mangabeys developing at least one visible skin lesion and four deaths. Trained veterinarians wearing a complete set of personal protective equipment and following strict biosafety protocols[25] performed on-site necropsies on three of the four infants. The body of the fourth infant was never found.

To confirm infection with MPXV, we first tested necropsy samples from the three infants and identified viral DNA in all main organs (Supplementary Table 1). We then performed a group-wide outbreak investigation by analysing 170 faecal samples collected from the mangabeys during the outbreak window, defined as the period in which clinical signs were visible in the group (Extended Data Table 2 and Supplementary Table 2). We detected MPXV DNA in 36 faecal samples collected from 19 individuals (7 symptomatic and 12 asymptomatic; Fig. 3). Of these 19, 14 were mothers of symptomatic infants and only 6 of them developed lesions (Supplementary Table 3a,b). We did not detect MPXV in 89 faecal samples collected after clinical signs resolved. These findings show that MPXV caused disease in a large proportion of this group and may have infected an even larger pool of individuals subclinically, consistent with earlier observations in chimpanzees from TNP[21].

To assemble viral genomes and determine their relationships with MPXVs that previously emerged in this area, we applied hybridization capture coupled to high-throughput sequencing to both necropsy and faecal samples. This allowed us to assemble two near-complete genomes, from a skin sample collected on 31 January 2023 (371× average depth of coverage; 91.7% of the reference genome covered by at least 20 reads) and a faecal sample collected on 12 February 2023 (22×; 91.6% of the reference genome covered by at least five reads) (Extended Data Figs. 3 and 4). The two genomes were identical across their 180,606 overlapping positions (91.07% of the reference genome). We also generated partial genomic information from necropsy samples of another individual (2.9× from merged reads of liver and spleen samples; 22.2% of the reference genome covered by at least five reads), whose sequences were identical across its regions of overlap with higher quality genomes (Supplementary Information and Extended Data Fig. 3). Maximum-likelihood phylogenetic analyses placed this virus in the genetic diversity of clade IIa viruses, as close relative to the other MPXVs from TNP (Fig. 4). This suggests that this outbreak was the result of a transmission event involving the same local reservoir(s).

To investigate whether squirrels or other small terrestrial mammals could be a source of MPXV infection for the TNP NHP, we tested rodents and shrews trapped (*n* = 694) or found dead (*n* = 10) inside and around TNP between 2019 and 2024 (Supplementary Tables 4a,b and 5). We identified one MPXV-positive fire-footed rope squirrel found dead on 3 November 2022—12 weeks before the onset of the outbreak in the mangabeys and approximately 3 km south of their territory (Extended Data Fig. 5). All organs from the squirrel (*n* = 15), as well as oral and nasal swabs, contained high viral loads and we were able to isolate viable MPXV from the skin, lung, spleen and liver of the animal (Supplementary Table 5). Although the cause of death could not be determined and may therefore be the MPXV infection (necropsy report available in Supplementary Information), we note that many viruses, including the smallpox virus (*Orthopoxvirus variola*), are pathogenic in their natural host despite long-term co-evolution. In consequence, this detection is compatible with the hypothesis that rope squirrels are natural hosts of MPXV. Similarly, although we cannot exclude that this particular infection resulted from a cross-species transmission from another host, we believe that the growing body of historical, spatial and ecological evidence, including that presented here, supports the view that rope squirrels are a natural host of MPXV. We also sequenced the complete genome of this squirrel-infecting virus (114× from samples, 91.5% of the reference genome covered by at least 20 reads; 20× from the lung isolate, 81.9% of the reference genome covered by at least 20 reads) (Extended Data Fig. 3). Across 181,351 overlapping bases (91.44% of the reference genome), this genome was nearly identical to the MPXV genomes derived from the mangabey samples, except in four repetitive regions in which three small deletions and one insertion were observed (Supplementary Information). To interpret this near-identity (and identity when excluding insertion–deletion (indel) differences), we ran Bayesian phylogenetic analyses using molecular clock models.

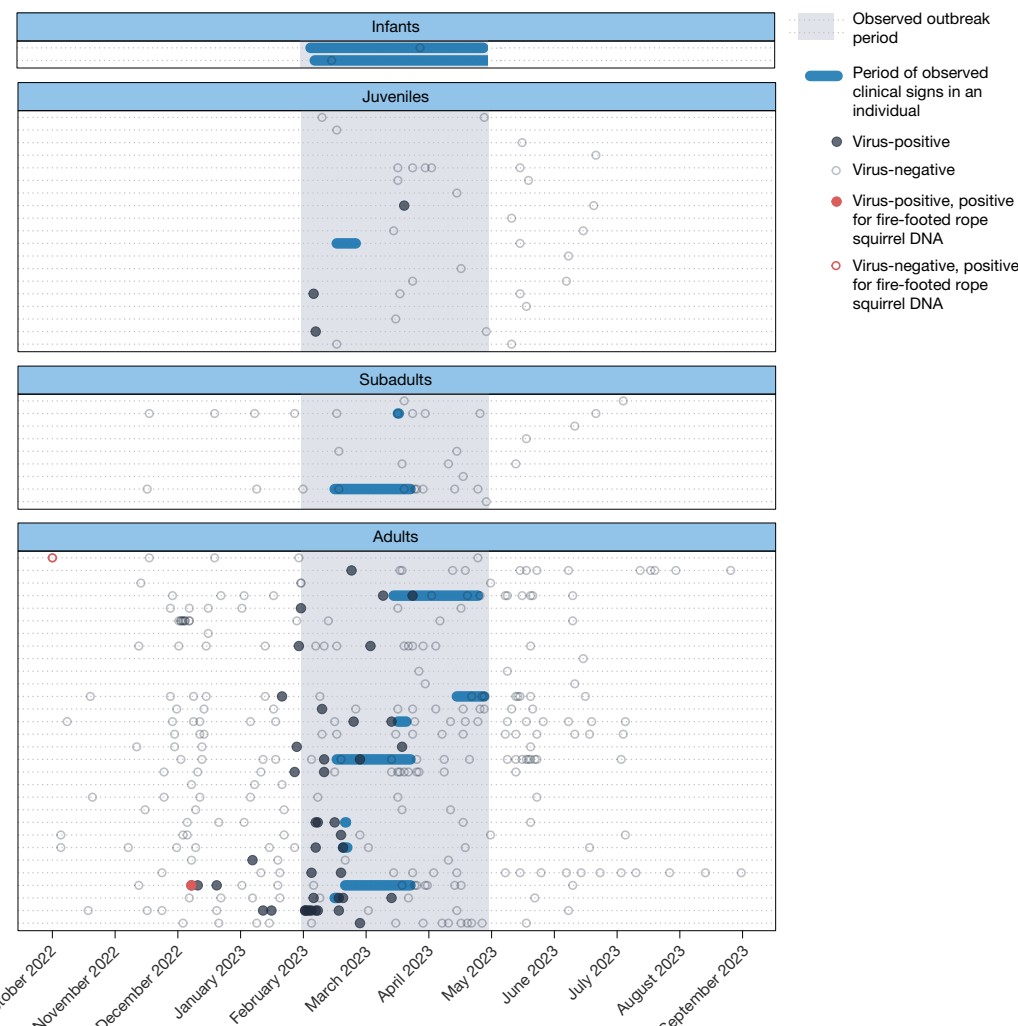

**Fig. 3 | Detection of MPXV DNA and of fire-footed rope squirrel DNA in sooty mangabey faeces.** Each line represents a different individual. Filled circles represent virus-positive samples, empty circles represent virus-negative samples. The solid red circle indicates the detection of squirrel DNA in a sample that was also MPXV positive and the unfilled red circle represents the detection of squirrel DNA in a sample that was MPXV negative. Vertical grey shadowing indicates the period in which clinical signs were observed in the mangabey group. For each individual, the blue horizontal shadowing indicates the period during which clinical signs were observed.

We obtained a substitution rate estimate of $3.0 \times 10^{-6}$ substitution per site per year (95% highest posterior density (HPD): $2.3 \times 10^{-6}$–$3.8 \times 10^{-6}$) under a strict clock model and $4.5 \times 10^{-6}$ substitution per site per year (95% HPD: $2.2 \times 10^{-6}$–$7.4 \times 10^{-6}$) under an uncorrelated log-normal relaxed clock model, slightly higher than previous estimates[21]. Under these rates, it may take up an average 13 (relaxed clock) or 20 months (strict clock) before two epidemiologically linked sequences diverge at a single genomic site (excluding indels, whose evolution is not modelled here). In line with this expectation, the time to the most recent common ancestor of the mangabey and squirrel viruses was dated to 2021 (95% HPD: 2019–2022) (Extended Data Fig. 6). However, if the events that led to the squirrel infection and the mpox outbreak in sooty mangabeys were epidemiologically related, one would also expect the two viral genomes to be identical (excluding indels). In any case, these genomic data can only point at recent events of cross-species transmission. We hypothesized that the most likely of these events was the transmission of MPXV from fire-footed rope squirrels to mangabeys.

To strengthen this hypothesis, we then aimed at refining a plausible scenario of MPXV emergence. For this, we first tested 114 faecal samples collected from the mangabey group in the 16 weeks before the outbreak. We identified 10 of 114 MPXV-positive faecal samples (detection rate 8.8%; 95% confidence interval 4.8–15.9). MPXV DNA was present in

faecal samples of seven mangabeys that were asymptomatic at the time of detection, including the mother of the first infant to show clinical signs (Fig. 3). Importantly, the three earliest positive samples were consecutively obtained from the same individual, on 6, 9 and 18 December 2022. Collectively, these results indicate that MPXV entered the group by means of this plausible index case and then circulated undetected in the group for nearly 2 months.

Mangabeys are known to hunt small mammals, including in TNP. Reviewing available long-term behavioural data, we found a video recording from 2025 showing a mangabey catching a squirrel (Supplementary Video 1) and an older video from 2014 showing a mangabey feeding on a clearly identifiable fire-footed rope squirrel (Fig. 5 and Supplementary Video 2). To explore whether a fire-footed rope squirrel hunt may have been the source of this outbreak, we analysed the mangabey diet before the outbreak by searching for prey DNA in the 78 earliest faecal samples of our collection (Supplementary Table 2). Using mammal-generic metabarcoding, we identified DNA sequences perfectly matching the mitogenome of the TNP fire-footed rope squirrel in two faecal samples (Supplementary Information). This demonstrates that these squirrels are not only part of the mangabey diet, but that group members fed on this species on at least two distinct occasions in the weeks before the outbreak. Even more

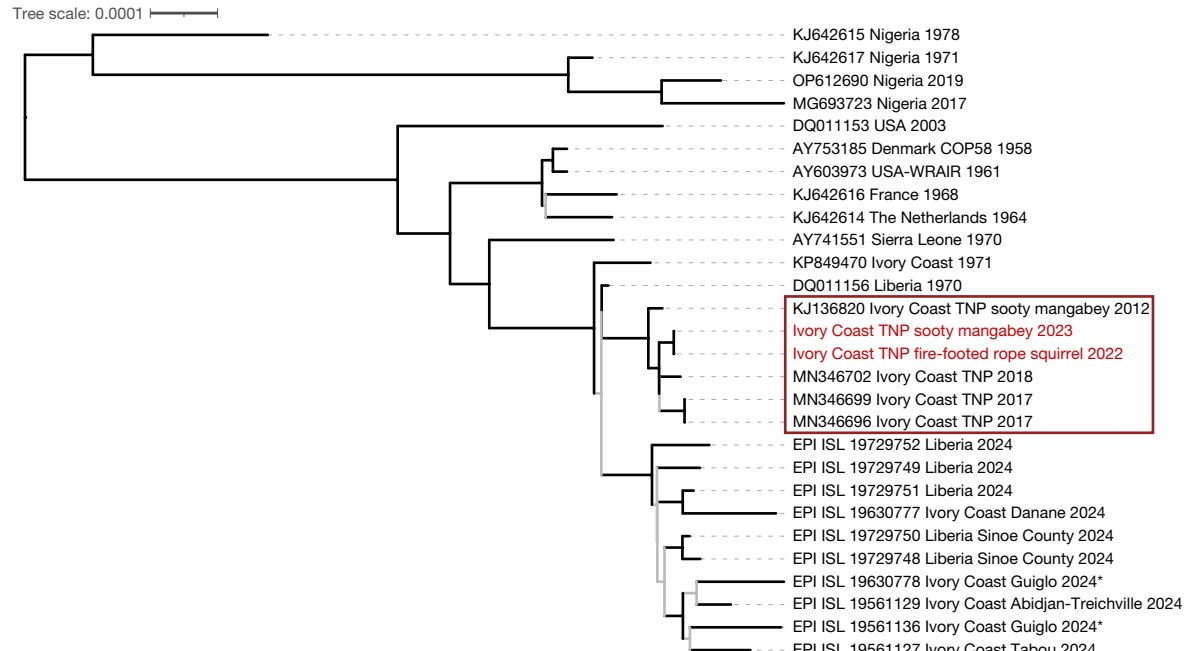

Tree scale: 0.0001

KJ642615 Nigeria 1978
KJ642617 Nigeria 1971
OP612690 Nigeria 2019
MG693723 Nigeria 2017
DQ011153 USA 2003
AY753185 Denmark COP58 1958
AY603973 USA-WRAIR 1961
KJ642616 France 1968
KJ642614 The Netherlands 1964
AY741551 Sierra Leone 1970
KP849470 Ivory Coast 1971
DQ011156 Liberia 1970
KJ136820 Ivory Coast TNP sooty mangabey 2012
Ivory Coast TNP sooty mangabey 2023
Ivory Coast TNP fire-footed rope squirrel 2022
MN346702 Ivory Coast TNP 2018
MN346699 Ivory Coast TNP 2017
MN346696 Ivory Coast TNP 2017
EPI ISL 19729752 Liberia 2024
EPI ISL 19729749 Liberia 2024
EPI ISL 19729751 Liberia 2024
EPI ISL 19630777 Ivory Coast Danane 2024
EPI ISL 19729750 Liberia Sinoe County 2024
EPI ISL 19729748 Liberia Sinoe County 2024
EPI ISL 19630778 Ivory Coast Guiglo 2024*
EPI ISL 19561129 Ivory Coast Abidjan-Treichville 2024
EPI ISL 19561136 Ivory Coast Guiglo 2024*
EPI ISL 19561127 Ivory Coast Tabou 2024

**Fig. 4 | Maximum likelihood phylogeny of clade IIa MPXVs.** The red box highlights MPXV genomes sampled in TNP. Genomes retrieved from the sooty mangabey outbreak and the fire-footed rope squirrel are shown in red. Asterisks indicate sequences obtained from human mpox cases in the neighbouring town of Guiglo in Côte d'Ivoire. The scale bar represents substitutions per variable site. Shimodaira–Hasegawa-like likelihood ratio test values of the inner branches are indicated by colour (grey, less than 0.90; black, equal to or more than 0.90).

striking, we found that one of the two samples containing squirrel DNA was also the first MPXV-positive faecal sample of the suspected index case (Fig. 3). The codetection of squirrel and MPXV DNA in this faecal sample strongly hints at an exceptional case of real-time detection of a cross-species transmission event, which subsequently led to the group-wide MPXV outbreak. Because the squirrel detected in this faecal sample cannot be the same as the one found dead more than a month earlier well outside the mangabey home range (Fig. 1), this codetection of MPXV and squirrel DNA constitutes further independent proof of MPXV circulation in fire-footed rope squirrels in the area during this period. Although our findings do not show per se that MPXV permanently circulates in this species, we believe that several lines of evidence now indicate that they are probably natural hosts[5,7–9]. Combined with the evidence of direct transmission reported in this study, the hypothesis that they serve as a reservoir of MPXV for wild NHP in TNP now seems plausible.

Bushmeat remains an important source of protein in sub-Saharan Africa, including Côte d'Ivoire. In many regions, the recent decline of large-bodied mammals due to habitat destruction and hunting has induced a shift in consumption towards smaller animals, especially rodents[26–29]. A study conducted in the villages bordering TNP has shown that, although primates remain the most hunted taxa, rodents are also commonly traded and consumed[10]. Although larger rodents, such as the giant pouched rat (*Cricetomys* sp.) and the marsh cane rat (*Thryonomys swinderianus*), are more frequently seen on markets, several species of squirrels are also sold and consumed, in both rural and urban areas[11]. It is also worth noting that, contrary to NHP who mainly rely on intact forest ecosystems, squirrels can thrive in fragmented habitats and plantations close to villages[30]. On re-analysing a mammal-generic metabarcoding dataset derived from carrion flies collected along a gradient from pristine forest to the surrounding villages at the same site[31], we only detected fire-footed rope squirrels in secondary forests and plantations, suggesting a higher presence of this species in these habitats (Supplementary Table 6). In such areas, squirrels are commonly trapped by the local population, including children, and directly

consumed, creating opportunities for zoonotic transmissions. Both subsistence hunting and bushmeat hunting, trade and consumption may result in MPXV transmission to humans.

Research on the ecology, habitat use and population dynamics of fire-footed rope squirrels, as well as the dynamics of MPXV infections in these populations and their interactions with humans, will be key to assessing spillover risks from this species. Efforts aimed at identifying other small mammal species that may serve as natural hosts and reservoirs should also be continued, because the involvement of several host species seems likely[8,17]. At the same time, MPXV genomic surveillance in humans in endemic areas remains the most abundant source of information on the diversity of this virus in its reservoir(s). For example, we observed that genomes published from human mpox

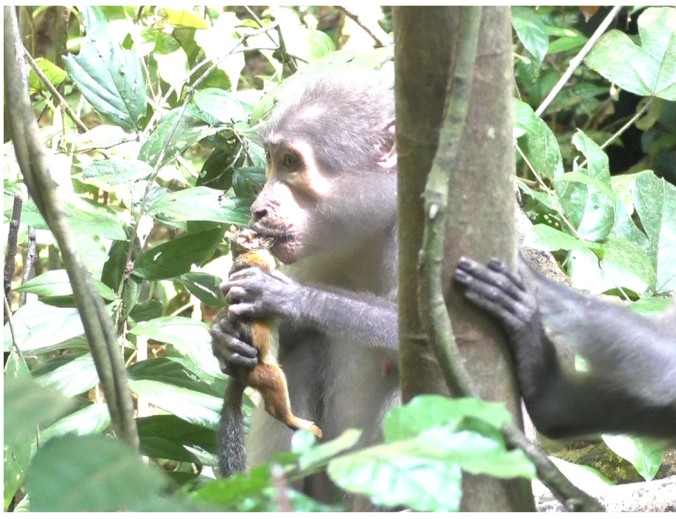

**Fig. 5 | Sooty mangabey eating a fire-footed rope squirrel in TNP.** This adult female was observed eating a squirrel on 9 December 2014. Photo by Taï Chimpanzee Project/Alexander Mielke.

cases caused by clade IIa MPXVs in Côte d'Ivoire in 2024, including two from a town about 80 km north of our study site, were not closely related to the ones circulating in TNP wildlife (Fig. 4). This confirms a large strain diversity, even at small spatial scales and suggests geographic structure and evolution in distinct hosts, although further studies are needed to test this. A better understanding of MPXV ecology will inform local authorities in charge of public health, animal health, as well as protected areas and natural resources, and help them to develop programmes to assess and mitigate spillover risk. This might include campaigns to raise awareness about the general risks linked to bushmeat and initiatives to codesign measures specifically aimed at reducing contact with squirrels through subsistence hunting. The example of TNP simultaneously shows the direct link between a rodent host and a spillover host and that both may be sources of human infections, suggesting that a focus on squirrel consumption alone would be misguided.

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

# Methods

## Health monitoring and sampling

TNP is the largest remaining primary rainforest in Western Africa. Its wild populations of NHP have been studied by the TCP since 1979 (ref. 34). TCP established a veterinary programme from 2001 onwards[13]. This veterinary programme conducts wildlife mortality surveillance and health monitoring of the four neighbouring groups of chimpanzees and one group of sooty mangabeys that are habituated to human observers. The mangabey group (named the Audrenisrou group) was habituated in November 2012 (ref. 35), and at the time of the outbreak consisted of about 80 individuals. The habituated groups are followed daily by trained field assistants and research staff. Names are given to each habituated individual. Newborns are given a temporary name indicating that they are the infant (BB) of a certain individual (for example, BB-Atacama indicates the newborn of Atacama). Behavioural data, as well as faecal and urine samples, are routinely collected from all the adults of the group. Faecal samples are collected with a plastic spatula right after defecation occurs and stored in 2-ml cryotubes. Urine is collected with fine Pasteur pipettes from underlying vegetation as soon as the animals urinate from a higher position, and is stored in 2-ml cryotubes. These samples are preserved in liquid nitrogen in the field, transported to Germany in dry ice and then stored at −80 °C until further analysis. When clinical signs are observed in the groups, observations and sampling are intensified. During this MPXV outbreak faecal samples were collected in both dry 2-ml cryotubes and in cryotubes containing nucleic acid preserving (NAP) buffer from most individuals of the group belonging to all age categories (infants, juveniles, sub-adults and adults), and from both symptomatic and asymptomatic individuals. Faecal samples are difficult to collect from infants, therefore the number of these samples is lower than other age classes. It is also important to note that in 2022 a substantial number of male juveniles immigrated to the Audrenisrou group, and many births occurred, leading to an increase in the total population to 80 individuals. To obtain an overview of viral DNA shedding in the mangabey group, we tested faecal samples from three key time periods: from 4 months before the first observations of clinical signs (1 October 2022 to 26 January 2023), during the outbreak (27 January 2023 to 26 April 2023) and up to 4 months after the last symptoms were observed (27 April 2023 to 24 August 2023). A total of 444 faecal samples were tested for MPXV, including those from just before ($n = 114$), during ($n = 170$) and after ($n = 89$) the mpox outbreak in the mangabey group. Details are provided in Supplementary Table 2. The veterinary team of TCP also performs necropsies on all animals found dead in the research area. Necropsies are done by trained veterinarians wearing full personal protective equipment. All used materials are incinerated or disinfected with 1% sodium hypochlorite solution and the carcasses are buried, according to the WHO guidelines. Samples are collected from all inner organs when carcass decomposition is not too advanced and stored in 2-ml cryotubes, both empty and filled with NAP buffer. The cryotubes are then preserved in liquid nitrogen in the field, transported to Germany in dry ice and stored at −80 °C until further analysis. In this study, we included 88 necropsy samples from 23 carcasses representing 11 species (and 4 species for which taxonomic assignment was not possible) collected between 2019 and 2024. Further details are provided in Supplementary Tables 1 and 5.

## Trapping of small terrestrial mammals inside and around TNP

Rodents and shrews were trapped using Sherman, Havahart-style or 0.5-m cage traps and were anaesthetized using a combination of ketamine (mouse dose 50 mg kg⁻¹, rats dose 35 mg kg⁻¹; Medistar) and xylazine (mouse dose 5 mg kg⁻¹, rat dose 3.5 mg kg⁻¹; WDT) intramuscularly. After being anaesthetized, the animals were measured, weighed and sampled. After sampling, the animals were marked and placed in an individual box until full recovery was observed. Anaesthetized animals were monitored closely and in some cases antisedan (5 mg kg⁻¹; Vetoquinol) was administered intramuscularly to facilitate recovery. Saline solution was sometimes applied as a subcutaneous infusion to prevent dehydration and applied on the eyes to prevent from drying. After recovery, the animals were released where they were caught. From July to November 2021 and March to April 2023, 173 rodents were trapped in the territory of the mangabey group. From these two field missions, we collected 167 oral swabs, 167 rectal swabs, 23 nasal swabs, 133 faecal samples and 39 samples from skin lesions. Eight individuals died in the trap or succumbed to anaesthetization, and one was euthanized because of signs of extreme weakness. In these cases, necropsies were performed and samples were collected from all main organs. All samples were stored in 2-ml cryotubes dry or with NAP buffer, frozen in liquid nitrogen in the field, transported in dry ice to Germany and then kept at −80 °C. From these trapping missions, a total of 553 samples, including different tissues and swabs were tested for OPVs. We also made use of samples originating from a broader initiative aimed at characterizing the biodiversity of small mammals and related pathogens along a gradient spanning from three villages bordering TNP on the west to the pristine forest in the immediate vicinity of the sooty mangabey territory. We set traps along three parallel transects of about 9 km covering distinct environments: (1) anthropic/domestic (inside houses), (2) at the village periphery, (3) at the edge between cultivated fields and the national park and (4) in the pristine forest of the national park. Sampling was performed from July to September 2021 and April to May 2022. In total, 521 rodents and shrews were trapped and sampled (as mentioned above), of which 82 were euthanised and underwent a full necropsy. Oral and rectal swabs were stored in 2-ml tubes with NAP buffer at room temperature until transport to Abidjan where they were stored at −20 °C. Necropsy samples were stored in 2-ml cryotubes and immediately frozen in liquid nitrogen. Samples were transported to Germany on dry ice and then stored at −80 °C (necropsies) or −20 °C (swabs in NAP buffer). From this sample set, we tested 506 oral swabs, 269 rectal swabs and different organs from the 82 necropsies. In toto, 1,011 samples from different tissues and swabs were tested for OPV. Details for the sampled animals are provided in Supplementary Table 4a,b.

## DNA extraction and OPV DNA detection

Nucleic acids were extracted from 40 mg of faecal matter using the GeneMATRIX Stool DNA Purification Kit (Roboklon). For the necropsy samples, 20 mg of tissue were used for DNA extraction with the DNeasy Blood and Tissue kit (Qiagen) or QIAamp Viral RNA Mini Kit (Qiagen). Nucleic acids from virus isolates were extracted using the NucleoMag VET Kit (Macherey-Nagel) and with the RNAdvance Tissue Kit (Beckman Coulter). DNA extracted from faecal and necropsy samples (excluding rodents) was tested for MPXV in duplicate using a TaqMan real-time quantitative PCR (qPCR) targeting the G2R locus[36]. Each PCR reaction had a total volume of 25 µl and included the following components: 5 µl of DNA template, 11.8 µl of nuclease-free water, 2.5 µl of 10× reaction buffer, 2 µl of 50 mM MgCl₂, 1 µl of 2.5 mM dUTPs, 1 µl of 10 µM G2R G forward primer (5′-GGAAAATG TAAAGACAACGAATACAG-3′), 1 µl of 10 µM G2R G reverse primer (5′-GCTATCACATAATCTGGAAGCGTA-3′), 0.5 µl of 10 µM G2R G probe (AAGCCGTAATCTATGTTGTCTATCGTGTCC) and 0.2 µl of Platinum Taq polymerase. PCR cycling conditions consisted of an initial denaturation at 95 °C for 6 min, followed by 45 cycles of 95 °C for 5 s and 60 °C for 30 s. Rodent DNA extracts (including the DNA extracts from trapped rodents and necropsies) were tested in duplicate for OPV using a TaqMan real-time PCR targeting the *P4A* gene[37]. Each reaction was prepared in a total volume of 25 µl, consisting of 5 µl of DNA template, 12.7 µl of nuclease-free water, 2.5 µl of 10× reaction buffer, 2 µl of 50 mM MgCl₂, 1 µl of 2.5 mM dUTPs, 0.75 µl of 10 µM OPV forward primer (TAATACTTCGATTgCTCATCCAGG), 0.75 µl of 10 µM OPV reverse primer (ACTTCTCACAAATGGATTTGAAAATC), 0.1 µl of 10 µM OPV TMgB probe (6FAM-TCCTTTACGTG+A + T + A + A + A + T + C + A + T)

and 0.2 μl of Platinum Taq polymerase. PCR cycling conditions were set to an initial denaturation at 95 °C for 10 min and 45 cycles of 95 °C for 15 s and 60 °C for 34 s. Positive extracts were then tested with the MPXV-specific qPCR mentioned above. A confirmatory PCR targeting a 270 base pair (bp) fragment of the haemagglutinin (*HA*) gene of OPVs[38] was performed for all the extracts that had weakly positive results in the MPXV or OPV qPCRs. For this assay, a single reaction had a total volume of 25 μl, containing 5 μl of DNA template, 11.8 μl of nuclease-free water, 2.5 μl of 10× reaction buffer, 2 μl of 50 mM MgCl₂, 2 μl of 2.5 mM dUTPs, 0.75 μl of 10 μM OPV.HA-156 forward primer (GGAGCCCAATTCCATT ATTC), 0.75 μl of 10 μM OPV.HA-424 reverse primer (gTATTATgTCT ATAgTCgATTCACTATCTg) and 0.2 μl of Platinum Taq polymerase. The PCR protocol included an initial denaturation at 95 °C for 5 min, followed by 45 cycles of 95 °C for 15 s, 60 °C for 30 s and 72 °C for 60 s, with a final extension at 72 °C for 7 min and a hold at 4 °C. The PCR products were then visualized by electrophoresis on a 2% agarose gel.

## Mammal species identification

For molecular species identification, two PCR systems targeting the mitochondrial genome were used. The first system designed by Geller and colleagues[39] targets the *CO1* gene. Each reaction contained 2.5 μl of DNA template, 14.8 μl of nuclease-free water, 2.5 μl of 10× reaction buffer, 1 μl of 50 mM MgCl₂, 1 μl of 2.5 mM dUTPs, 1 μl of BSA (1 mg ml⁻¹), 1 μl of 10 μM forward primer jgLCO1490 (5′-TITCIACIAAYCAYAARGAYA TTGG-3′), 1 μl of 10 μM reverse primer jgHCO2198 (5′-TAIACYTCIGGRTG ICCRAARAAYCA-3′) and 0.2 μl of Platinum Taq polymerase. The PCR protocol included an initial denaturation at 94 °C for 2 min, followed by 47 cycles of 95 °C for 30 s, 52 °C for 30 s and 72 °C for 50 s, with a final extension at 72 °C for 2 min and a hold at 8 °C. The PCR products were then visualized by electrophoresis on a 1.5% agarose gel. The second system targets the cytB gene[40]. Each reaction contained 1 μl of DNA template, 16.25 μl of nuclease-free water, 2.5 μl of 10× reaction buffer, 2 μl of 50 mM MgCl₂, 2 μl of 2.5 mM dUTPs, 0.5 μl of 10 μM forward primer CytB-outF (5′-CGAAGCTTGATATGAAAAACCATCGTTG-3′), 0.5 μl of 10 μM reverse primer CytB-inR (5′-AGTGGRTTRGCTGGTGTRTARTTG TC-3′) and 0.25 μl of Platinum Taq polymerase. The PCR protocol included an initial denaturation at 95 °C for 5 min, followed by 40 cycles of 95 °C for 30 s, 52 °C for 30 s and 72 °C for 45 s, with a final extension at 72 °C for 10 min and a hold at 8 °C. The PCR products were then visualized by electrophoresis on a 1.5% agarose gel. If a band was visible at the target lengths of the PCRs, the PCR product was Sanger sequenced. After removal of the primer target-regions in Geneious Prime 2025.1.2 (https://www.geneious.com), a query search of the resulting reads to identify the best sequence matches was performed on Nucleotide BLAST (https://blast.ncbi.nlm.nih.gov/Blast.cgi). If molecular identification of species failed, the animals were determined morphologically following *Kingdon Field Guide to African Mammals*[41] and *Mammals of Africa* (Vol. III)[33].

## Virus isolation

Virus isolation was attempted from 13 faecal samples (12 from the mangabeys, 1 from the fire-footed rope squirrel), 13 tissue samples and maggots from two necropsies. Skin, lung and spleen were tested for each mangabey necropsy, as well as a maggot from one individual. The squirrel samples tested encompassed skin, lung, spleen, liver, faeces and maggots. The samples were added to cell culture medium with 10% fetal bovine serum supplemented with penicillin/streptomycin (Gibco) and gentamicin/amphotericin (Gibco), bead homogenized on a bead ruptor and incubated overnight at 8 °C. Sample homogenate was filtered through a 0.8-μm pore membrane to remove larger particles and potential contaminating bacteria. The filtrate was added to confluent layers of MA-104 cells and cultivated with the aforementioned antibiotic-supplemented medium in 12.5-cm² rectangular canted neck cell culture flasks. MA-104 cells originated from the Collection of Cell Lines in Veterinary Medicine, Insel Riems. The cell line has been authenticated by DNA barcoding of the cytochrome *b* gene, species-specific PCR, PCR targeting the aldolase gene and restriction fragment length polymorphism analysis. The cell line used in this study was not tested for mycoplasma contamination. Cell cultures were passaged after 3 days. If a cytopathic effect was visible, cells were passaged further to increase the viral titre for shotgun sequencing.

## Hybridization capture and high-throughput sequencing

Illumina-compatible dual index libraries were generated from up to 1,000 ng of DNA extracts from necropsies and four mangabey faecal samples (details in Supplementary Tables 1, 2 and 5). Faecal samples were selected on the basis of viral copy number. DNA was fragmented in 50 μl of low EDTA-TE buffer using a Covaris ME220 Focused-ultrasonicator (Covaris) set for a target fragment size of 350 bp (settings: treatment duration 45 s, peak power 50, duty factor 20%, 1,000 cycles per burst, average power 10, temperature 20 °C). Libraries were built from the fragmented DNA using the NEBNext Ultra II DNA kit following the manufacturer's recommendations. After the adaptor ligation, a 300–400-bp size selection using MagSi magnetic beads (Carl Roth) was performed if the input was higher than 50 ng of total DNA. Quantification of the final libraries was performed using the Kapa HiFi Library Quantification Kit (Roche) or the NEBNext Library Quant Kit for Illumina (New England Biolabs). Libraries were stored at −20 °C until further use. All libraries underwent MPXV target enrichment through in-solution hybridization capture with a previously described OPV bait set[10]. We used myBaits RNA baits following the myBaits sequence enrichment for targeted sequencing protocol (v.5.0; Daicel Arbor Biosciences) and applied two successive rounds of overnight (16–24 h) hybridization capture. Also, one round of overnight hybridization capture at 65 °C targeting the mitochondrial genome of rodents was performed on a library of the squirrel spleen. To design these custom baits, all complete mitochondrial genomes of rodents available on GenBank were accessed and redundancies were reduced by clustering genomes using CD-HIT30 at a minimum of 88% sequence identity. Final bait design was based on the resulting 239 accession numbers. For capture, only a quarter of the recommended bait quantity was used. Following each round of capture, the hybridized library pools were amplified using the KAPA HotStart Library Amplification Kit (Roche) to obtain a minimum of 200 ng total DNA per library pool. After final quantification using the Kapa HiFi Library Quantification Kit (Roche) or the NEBNext Library Quant Kit for Illumina, the enriched pools were diluted to the recommended concentrations. Sequencing was performed on a MiniSeq platform (Illumina) using the v.3 chemistry (MiniSeq High output Kit for 75 or 150 cycles). For whole-genome sequencing of MPXV from cell cultures, we generated libraries from isolates from two mangabey skin samples and from the squirrel lung using the Rapid-Barcoding Kit v.14 (Oxford Nanopore Technologies) and sequenced them directly on a PromethION 2 solo platform (Oxford Nanopore Technologies) using R10.4.1 PromethION flow cells. Basecaller v.4.3.0 was set to super-accurate basecalling v.4.3.0, 400 bp.

## Sequencing data analyses

Reads from different tissues of the same individual were merged to improve viral genome coverage. Raw sequencing reads were quality-filtered using trimmomatic v.0.39 (ref. 42) using the settings: LEADING:30 TRAILING:30 SLIDINGWINDOW:4:30 MINLEN:30. Filtered reads were then mapped to the most recent MPXV genome from TNP (GenBank accession number MN346702) using BWA MEM v.0.7.17-r1188 (ref. 43). Mapped reads were sorted and duplicates removed using SortSam and MarkDuplicates by Picard v.2.13.3 (http://broadinstitute.github.io/picard/). In parallel, paired reads were mapped to the reference genome in Geneious Prime 2023.1.2 (https://www.geneious.com) using default settings to improve coverage of inverted terminal repeat regions of the MPXV genome. Consensus sequences were generated from the reference-based mapping pipeline and the Geneious mapper

and checked manually for concordance. Criteria for consensus calling were set to a minimum unique read depth threshold of 20% and a 95% nucleotide frequency in the reads. If a nucleotide at any given position in the genome was found at a frequency less than 95%, an ambiguous base would be automatically called. For the faecal sample for which we obtained good coverage, but lower than the necropsy samples, consensus-calling criteria were set to a minimum unique read depth of 5% and a 50% nucleotide frequency in the reads. For remaining samples in which we obtained a shallow coverage, mapped reads were visually inspected in Geneious but no consensus sequence was called because of their low quality. Ambiguous bases and regions with a difficult read assembly (tandem repeats) were manually checked in consensus sequences of high-quality genomes. Nearly complete viral genomes (excluding the inverted terminal repeats) used for phylogenetic analyses were assembled from the reference-based mapping. The complete mitochondrial genome of the squirrel was de novo assembled from quality-filtered reads using SPAdes v.3.13.0 (ref. 44). Oxford Nanopore reads were quality trimmed using BBDuk Trimmer v.1.0 with the following settings: qtrim=rl trimq=6 minlength=50 ordered=t qin=33 (BBMap−Bushnell B.−sourceforge.net/projects/bbmap) and de novo assembled using Flye v.2.9.2 (ref. 45) (flags --nano-hq; --genome-size 200k). The entire dataset was then remapped against the initially generated sequence through Minimap2 v.2.17 (ref. 46) (ONT mode; including secondary alignments; maximum secondary alignments per read = 5; minimum secondary to primary alignment ratio = 0.8). Owing to data protection rules, all reads that could potentially be of human origin were removed before submission to the European Nucleotide Archive (BBDuk Trimmer v.1.0, mincovfraction=0.66, ref=GCF_000001405.40_GRCh38.p14_genomic.fna).

## Phylogenetic analyses

A dataset representing the current known MPXV clade IIa diversity was assembled from publicly available data on GenBank and GISAID[47]. For identical sequences originating from the same outbreak only one representative genome was selected. We also included partial genomes from Côte d'Ivoire and Liberia from 2024. After evaluating the Côte d'Ivoire sequences through Nextclade v.3.12.036 (ref. 48) quality control (https://clades.nextstrain.org/), we identified three high-quality genomes, which we included in our analysis. Furthermore, two genomes of lower quality were added because of their origin in geographic proximity to TNP. This dataset plus one representative MPXV genome per species from the TNP 2022/2023 outbreak ($n$ = 28) were aligned using MAFFT v.7.505n[49]. We used Squirrel v.1.2.2 (https://github.com/aineniamh/squirrel) to generate a masked alignment of 197,211 positions. We used this alignment to reconstruct a maximum-likelihood phylogeny using IQ-TREE v.2.1.4b[50,51]. Branch robustness was assessed by Shimodaira−Hasegawa-like approximate likelihood ratio tests[52]. We ran a regression of root-to-tip distances versus time and identified the best-fitting root of the resulting tree using TempEst v.1.5.3 (ref. 53) (Supplementary Fig. 9). For molecular clock analyses, we explored more finely the temporal signal in the tree using Phylostems[54]. The strongest temporal single was detected for a subtree of 24 sequences (Rsq = 0.71; Supplementary Table 7). We used this reduced dataset for further analyses with BEAST v.1.10.5, under strict and uncorrelated log-normal relaxed clock models[55]. We first validated the presence of a temporal signal with the Bayesian estimation of temporal signal (BETS) approach[56]. To do so, we compared marginal likelihoods estimates (MLE) of clock models with or without tip dates (in the second case, all tips are assumed to be contemporaneous). We ran several chains of all models and checked their mixing and convergence, as well as sufficient effective sample sizes of model parameters using Tracer v.1.7.2 (ref. 57) We found that the heterochronous model was decisively better than the isochronous one for both the strict ($MLE_{hetero}$: −256,462.8 versus $MLE_{iso}$: −256,505.1) and relaxed clock model (−258,458.7 versus −256,471.0), supporting the existence of a temporal signal in both cases and a better performance of the relaxed clock model. We summarized the posterior set of trees under the form of a maximum clade credibility tree using TreeAnnotator v.1.10.5 (distributed with BEAST). All trees were further visualized and edited in iTOL v.7.1 (https://itol.embl.de/).

## Diet analysis

The mangabey's diet was analysed using a metabarcoding approach. The faecal samples used in this particular study ($n$ = 78) were collected just before the mpox outbreak started in the mangabey group (1 October 2022–30 December 2022). We used the Tagsteady protocol[58]. A first PCR assay targeting a 130 bp fragment of the 16S mitochondrial DNA was performed with tagged 16S mam1 (5′-CGGTTGGGGTGACCTCGGA-3′) and 16S mam2 (5′-GCTGTTATCCCTAGGGTAACT-3′) primers to identify each sample individually. This PCR was performed with the addition of human blocking primer (16Smam_blkhum 5′-CGGTTGGGGCGACCTCGGAGCAGAACCC-3′) to reduce the amplification of contaminant human DNA. A total volume of 25 μl was used for each reaction, which included 2 μl of DNA template. The cycling parameters were set as follows: 95 °C for 10 min, 35 cycles of 95 °C for 12 s, 59 °C for 30 s, 70 °C for 25 s and 72 °C for 7 min. Subsequently, we generated three pools comprising all positive samples together with the positive and negative controls from the same PCR plate (three plates with one replicate each; 262 amplicons in total). After end repair, we indexed the pools by ligation with Illumina full-length Y-adaptors that carried dual matching indexes (P5–P7). The indexed pools were sequenced on an Illumina iSeq 100 System. To analyse the resulting reads, we first assembled a reference database from the EMBL collection of vertebrate sequences (downloaded on 10 July 2024) on which we performed an in silico PCR with the OBItools (v.3.0.1b21) ecopcr command, allowing up to three mismatches between the primer and the reference sequences. We sorted the reads generated from the diet analysis to their respective PCR replicate using their 5′ nucleotide tags using OBItools and removed primer sequences. Paired-end reads were then merged using the OBItools Illuminapairedend command keeping only reads with an alignment quality score of more than 0.8 and a length more than 80 bp. Sequence variants were then collapsed with the obiclean command, but retaining a count of their appearance in each PCR replicate. We then compared the resulting aligned reads with our reference database to try to assign taxons by using the OBItools ecotag command. To consider a wildlife species detection event genuine, we required that at least two of the three replicates contained at least two times the maximum number of reads assigned a taxonomy in the negative controls. We also used Geneious to competitively map the trimmed reads to the mitogenome we generated from the squirrel spleen, as well as a human (NC_012920), chimpanzee (KU308547) and mangabey (NC_028592) mitogenomes. To assess whether this short 16S fragment contained sufficient variation for unambiguous taxonomic assignment, we compared it to all publicly available 16S sequences from squirrel genera known to occur in Côte d'Ivoire. The fragment clearly distinguished fire-footed rope squirrels from all other genera and species (Supplementary Information and Supplementary Fig. 10).

## Re-analyses of fly data

To investigate the distribution of fire-footed rope squirrels and sooty mangabeys along a local ecological gradient, we reanalysed a recently published mammal metabarcoding dataset derived from fly DNA[31]. To do this, we applied the same bioinformatics approaches used for faecal diet analyses to the 100 datasets (25 from the forest, 50 from the edge and 25 from villages) produced by this study (https://doi.org/10.5281/zenodo.7688126).

## Reporting summary

Further information on research design is available in the Nature Portfolio Reporting Summary linked to this article.

## Data availability

Raw reads resulting from MPXV target enrichment in the samples, from the DNA metabarcoding and from the shotgun sequencing of the viral isolates are available in the European Nucleotide Archive under project accession number PRJEB90150, sample accession numbers ERS24827514–ERS24827784, ERS25063784–ERS25063790, ERS25115682–ERS25115684 and run accession numbers ERR15137312, ERR15137322–ERR15137323, ERR15137325–ERR15137330, ERR15137332–ERR15137334, ERR15137343–ERR15137588, ERR15172398–ERR15172401, ERR15390135 and ERR15391827–ERR15391828. Consensus sequences of MPXV high-quality genomes are available in Pathoplexus, accession numbers PP_0031VY3.1 (sooty mangabey faecal sample), PP_0031VZ1.1 (sooty mangabey necropsy) and PP_0031W1X.1 (fire-footed rope squirrel necropsy). More MPXV genomes included for phylogenetic analysis were retrieved from GenBank (accession numbers AY603973, AY741551, AY753185, DQ011153, DQ011156, KJ136820, KJ642614, KJ642615, KJ642616, KJ642617, KP849470, MG693723, MN346696, MN346699, MN346702, OP612690) and GISAID (accession numbers EPIISL19561127, EPIISL19561129, EPIISL19561136, EPIISL19630777, EPIISL19630778, EPIISL19729748–EPIISL19729752). Source data are provided with this paper.

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

**Acknowledgements** We thank the Ministère de l'Enseignement Supérieur et de la Recherche Scientifique, the Ministère de Eaux et Fôrets in Ivory Coast and the Office Ivoirien des Parcs et Réserves for permitting the study. We are grateful to the Centre Suisse de Recherches Scientifiques en Côte d'Ivoire and the staff members of the TCP for their support. We thank all primatologists, research assistants, field assistants and veterinarians who collected samples for the TCP over the years. We acknowledge the contribution of A. Mossoun, V. K. Kouamé, E. K. Diané, P. G. Hugues, A. K. Hermann, K. Y. Sie, M. Y. Koffi, M. Djê bi Tah, F. F. Logoué, K. A. Pléh, J. Steiner and O. Dimov to the small mammal trapping in TNP. We are grateful to M. D. Danzin for providing the video of the mangabey catching a squirrel in TNP and to Y. U. Ajamma, J. Gómez-Fortún and A. Graaf-Rau for their support with the mitotyping of the small mammal samples. This work was supported by the German Research Council Project LE1813/11-1 and LE1813/14-1 (Great Ape Health in Tropical Africa), the ARCUS Foundation grant G-PGM-2107-3516, the BIODIV-AFREID Project LE1813/17-1, the Heinrich Böll Stiftung PhD stipend to J.S. and the Evolution of Brain Connectivity Project of the Max Planck Society (M.IF.EVAN8103). Research was conducted under research permit numbers: 006/MESRS/DGRI (TCP 2022-2025), 461/MINEDD/OIPR/DT, 020/MESRS/DGRI and 007/ MESRS/DGRI (small terrestrial mammal trapping).

**Author contributions** Data and samples from the outbreak were collected by C.R.-F., A.L.-M. and the TCP field assistants and research assistants. Data and samples from the rodents in TNP were collected by L.L., L.K., J.S., M.J.S.J. and H.R.H. The field investigations as well as diagnostic and research activities were coordinated by S.C.-S., A.D., L.V.P. and F.H.L. Molecular laboratory analyses were performed by C.R.-F., J.S., H.R.H., L.L., A.D. and L.V.P. Diet analyses were performed by C.R.-F. and J.F.G. Virus isolation experiments and sequencing of the isolates were conducted by J.S., D.H., S.C. and coordinated by M.B. N.Y.N., H.K. and S.G.-B. provided information on bushmeat consumption in Côte d'Ivoire. R.M.W., C.C., L.S., A.D. and F.H.L. coordinated the mangabey fieldwork and provided the behavioural data. J.F.G., A.M. and R.M.W. provided the video. The data were analysed by C.R.-F., J.S., L.V.P. and S.C.-S. and the paper was drafted by C.R.-F., J.S., S.C.-S., L.V.P and F.H.L. The paper was revised and approved by all authors.

**Funding** Open access funding provided by Helmholtz-Zentrum für Infektionsforschung GmbH (HZI).

**Competing interests** The authors declare no competing interests.

**Additional information**
**Correspondence and requests for materials** should be addressed to Livia V. Patrono or Fabian H. Leendertz.

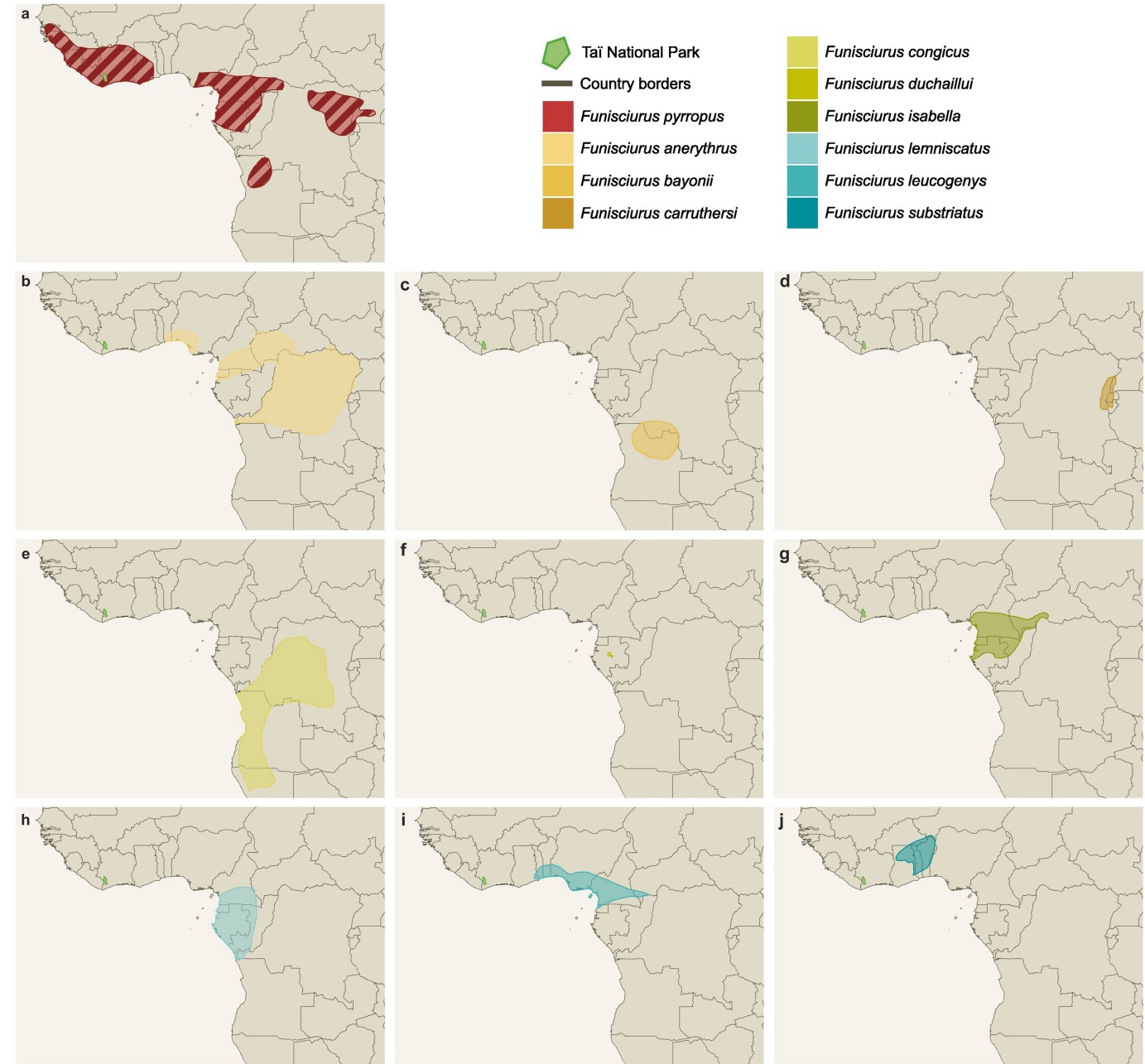

**Extended Data Fig. 1 | Home range of the different rope squirrel (*Funisciurus* sp.) species. a** *F. pyrropus*[32], **b** *F. anerythrus*[32], **c** *F. bayonii*[59], **d** *F. carruthersi*[60], **e** *F. congicus*[61], **f** *F. duchaillui*[62], **g** *F. isabella*[63], **h** *F. lemniscatus*[64], **i** *F. leucogenys*[65], **j** *F. substriatus*[66]. Map adapted from OpenStreetMap (https://www.openstreetmap.org), reproduced under a Creative Commons CC BY-SA 2.0 licence.

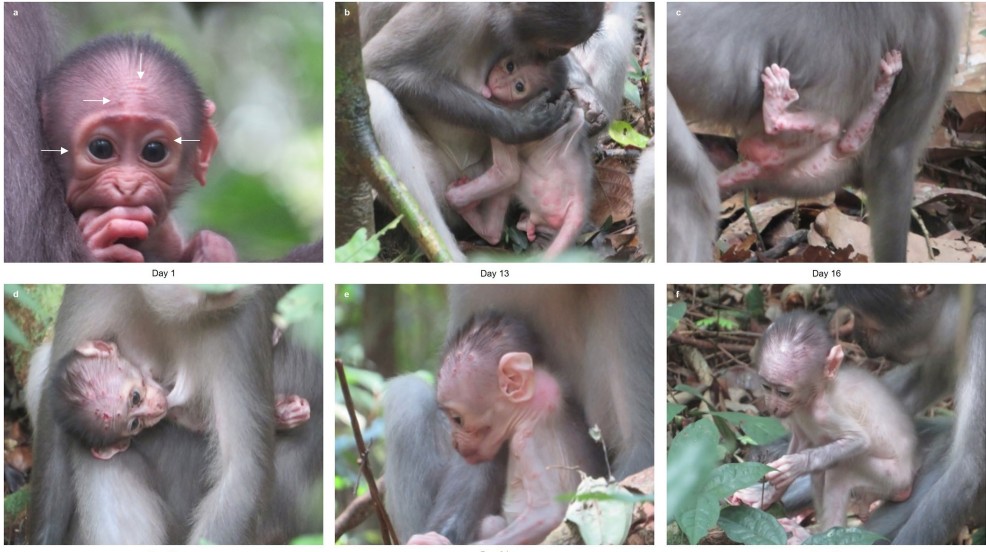

**Extended Data Fig. 2 | Evolution of MPXV-induced cutaneous rash in an infant of sooty mangabey (*Cercocebus atys*).** Day 1 refers to the first day that researchers observed symptoms in this individual. **a** A localized rash is present on the face (lesions are indicated by white arrows). **b** Multiple maculo-papular lesions appear in most parts of the body. **c** Some lesions have progressed to vesicles, and a few have advanced to pustules. **d** The lesions have formed crusts. **e** An enlarged lymphnode is visible on the left side of the neck. **f** The infant has recovered and scars are visible. Photo by Taï Chimpanzee Project/ Carme Riutord-Fe.

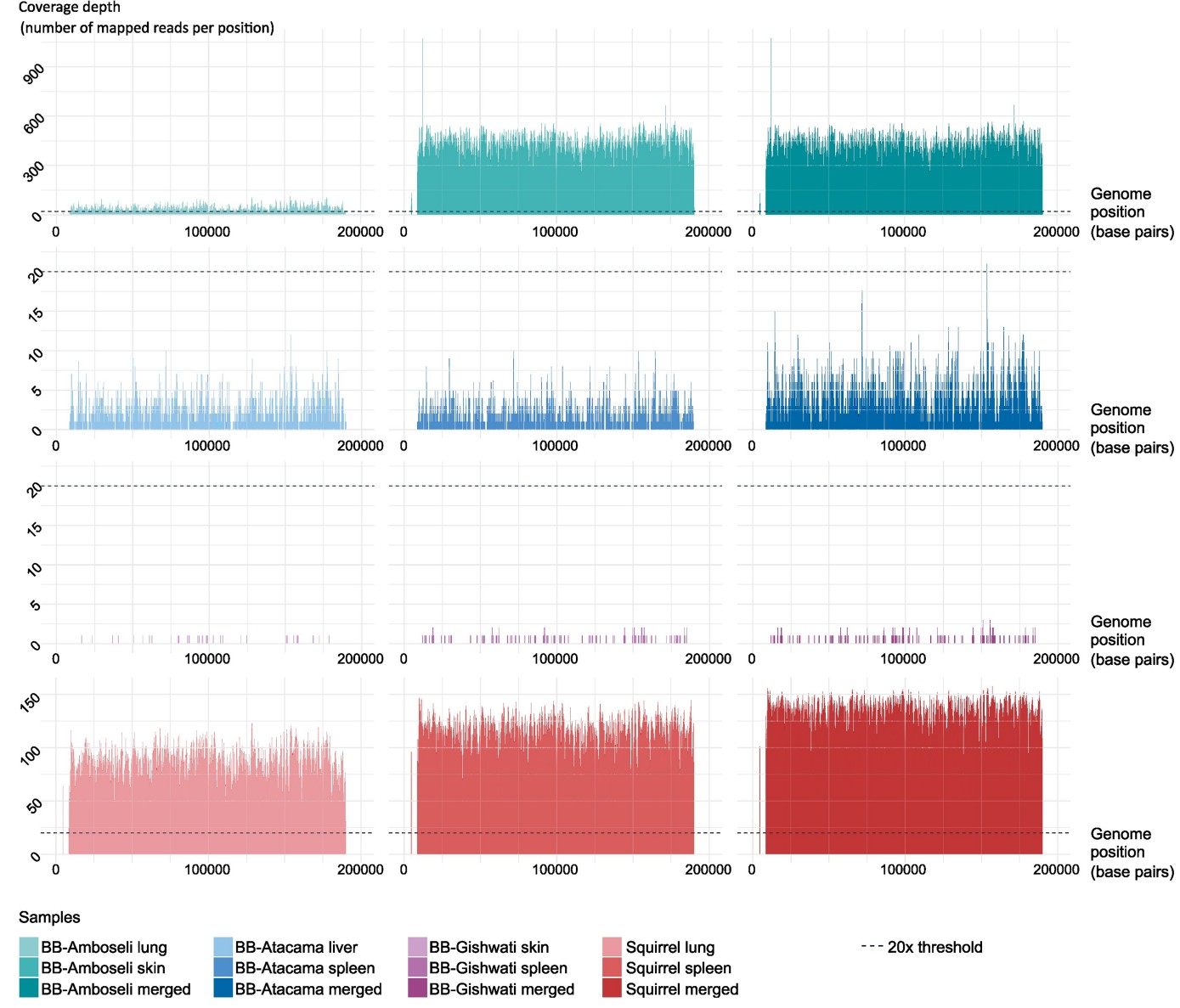

**Samples**

BB-Amboseli lung BB-Atacama liver BB-Gishwati skin Squirrel lung - - - 20x threshold

BB-Amboseli skin BB-Atacama spleen BB-Gishwati spleen Squirrel spleen

BB-Amboseli merged BB-Atacama merged BB-Gishwati merged Squirrel merged

**Extended Data Fig. 3 | Coverage plots for MPXV genomes assembled via hybridization capture and sequencing from necropsy samples of sooty mangabeys (*Cercocebus atys*) and a fire-footed rope squirrel (*Funisciurus pyrropus*).** Shown are the coverage plots representing MPXV genome coverage in the different tissue types of sooty mangabey and fire-footed rope squirrel necropsies. Each colour represents a different individual. Colour nuances represent different sample types from the same individual and merged data from the different samples.

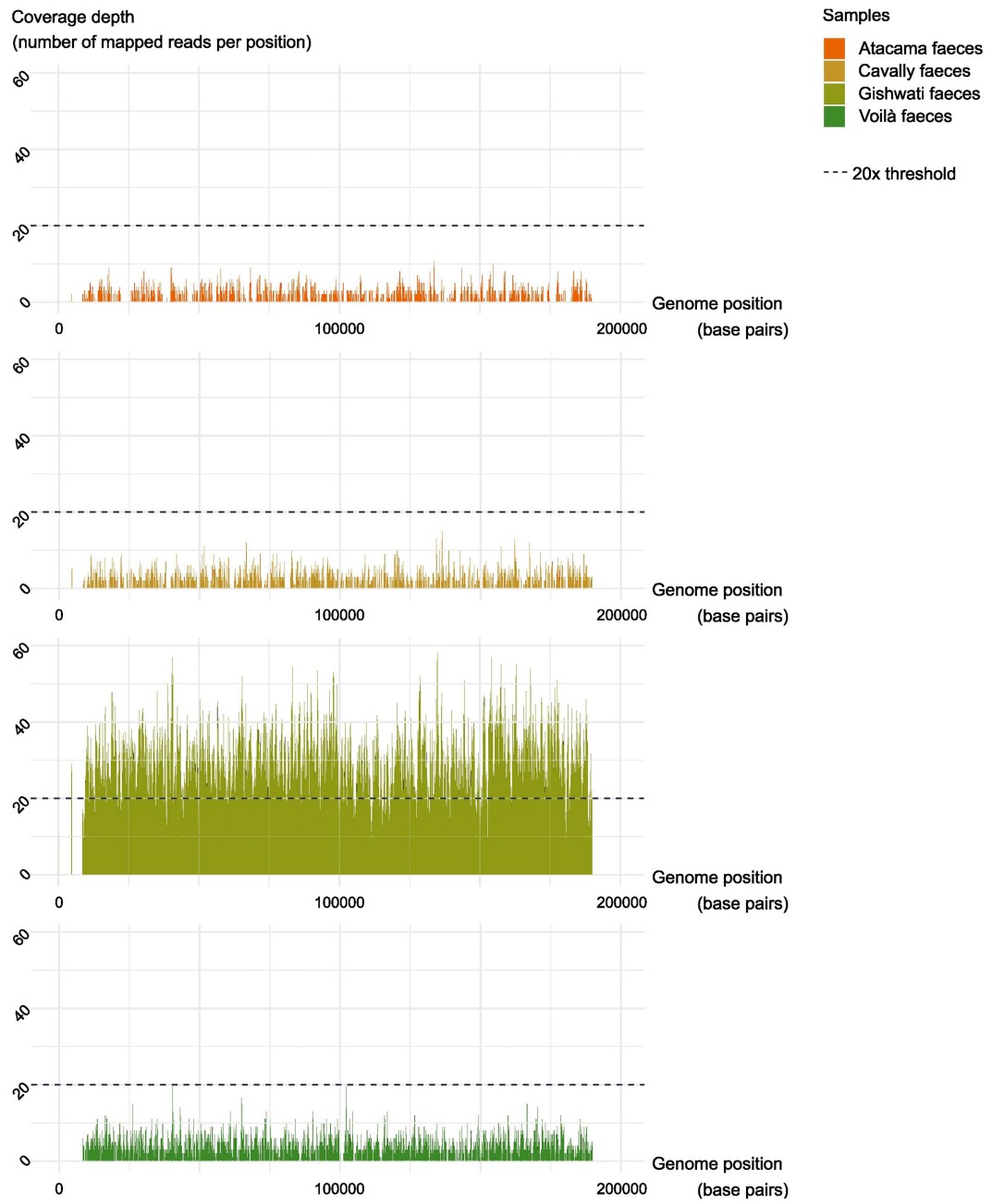

**Extended Data Fig. 4 | Coverage plots for MPXV genomes assembled via hybridization capture and sequencing from sooty mangabey (*Cercocebus atys*) faecal samples.** Shown are the coverage plots representing MPXV genome coverage obtained from the sooty mangabey faecal samples. Each colour represents a different individual.

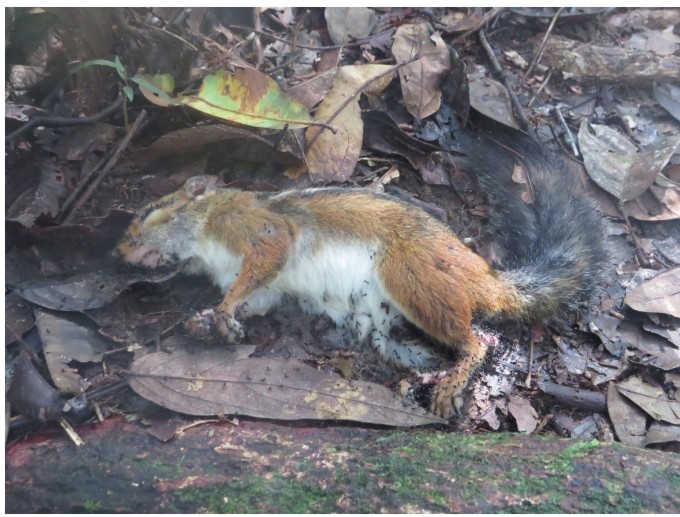

**Extended Data Fig. 5 | Carcass of the fire-footed rope squirrel (*Funisciurus pyrropus*) found dead in TNP.** Shown is the carcass of the fire-footed rope squirrel as it was found on November 3rd 2022 in TNP. Photo by Taï Chimpanzee Project/Carme Riutord-Fe.

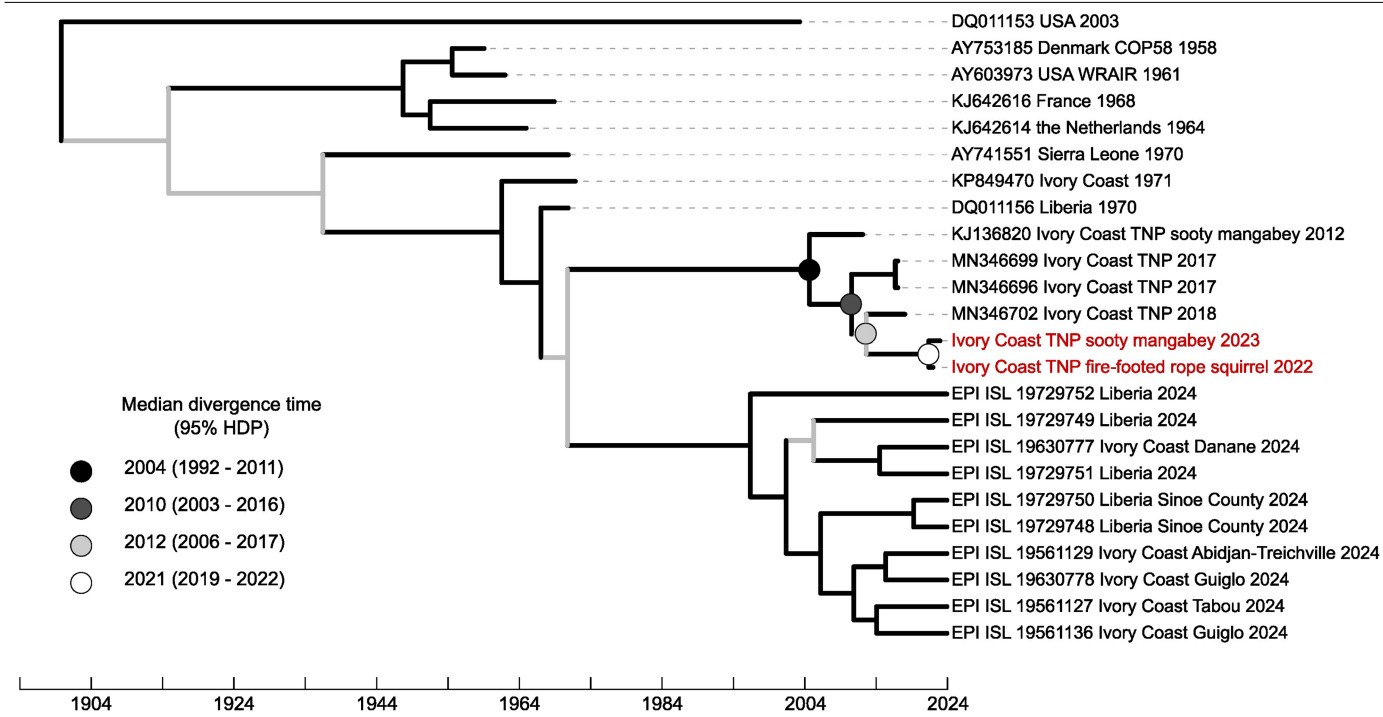

**Extended Data Fig. 6 | MPVX clade IIa time tree.** Viral genomes identified in a sooty mangabey (*Cercocebus atys*) and a fire-footed rope squirrel (*Funisciurus pyrropus*) in TNP are in red. The inner branch colours represent posterior probabilities (grey is <0.95; black is ≥0.95). The coloured nodes refer to estimates for the median divergence times and 95% HPD of the TNP MPXVs.

**Extended Data Table 1 | Degrees of rash severity associated with MPXV infection in the sooty mangabey (*Cercocebus atys*) group and MPXV DNA detection in asymptomatic individuals**

| | Mild<br>< 5 lesions | Moderate<br>5-20 lesions | Severe<br>> 20 lesions | Asymptomatic<br>MPXV+ faeces |
|---|---|---|---|---|
| Infants (15) | 0 | 9 | 6 | 0 |
| Juveniles (22) | 1 | 0 | 0 | 3 |
| Subadults (13) | 2 | 0 | 0 | 0 |
| Adults (30) | 5 | 3 | 0 | 11 |

The numbers in brackets following the age categories indicate the number of individuals per age category. The numbers under each severity group correspond to the number of mangabeys that showed this symptomatology. The last column represents the number of asymptomatic individuals in which MPXV DNA was found in their faecal samples. 2 individuals disappeared/died before the outbreak started but their faecal samples have been included in the analysis (before the outbreak). Age classification was based on both physical characteristics and behavioral traits (Bande et al., unpublished data). Infants (<6 months), juveniles (females: 6 months - 3 years, males: 6 months – 6 years), subadults: (females: 3-5 years, males: 6-10 years) and adults: (females: >5 years, males >10 years).

**Extended Data Table 2 | Summary of faecal samples and MPXV DNA detection before, during and after the appearance of clinical signs associated with MPXV infection in the sooty mangabey (*Cercocebus atys*) group**

| | Total | MPXV+ (%) | Symptomatic (%) | Asymptomatic (%) |
|---|---|---|---|---|
| before (01.10.2022-26.01.2023) | 114 | 10 (8.77%) | - | 7 (100%) |
| during (27.01.2023-26.04.2023) | 170 | 36 (21.18%) | 6 (31.58%) | 13 (68.42%) |
| after (27.04.2023-24.08.2023) | 89 | 0 | - | - |

Livia V Patrono

## Reporting Summary

## Statistics

For all statistical analyses, confirm that the following items are present in the figure legend, table legend, main text, or Methods section.

| n/a | Confirmed | |
|---|---|---|
| ☐ | ☒ | The exact sample size (*n*) for each experimental group/condition, given as a discrete number and unit of measurement |
| ☐ | ☒ | A statement on whether measurements were taken from distinct samples or whether the same sample was measured repeatedly |
| ☒ | ☐ | The statistical test(s) used AND whether they are one- or two-sided *Only common tests should be described solely by name; describe more complex techniques in the Methods section.* |
| ☒ | ☐ | A description of all covariates tested |
| ☒ | ☐ | A description of any assumptions or corrections, such as tests of normality and adjustment for multiple comparisons |
| ☐ | ☒ | A full description of the statistical parameters including central tendency (e.g. means) or other basic estimates (e.g. regression coefficient) AND variation (e.g. standard deviation) or associated estimates of uncertainty (e.g. confidence intervals) |
| ☒ | ☐ | For null hypothesis testing, the test statistic (e.g. *F*, *t*, *r*) with confidence intervals, effect sizes, degrees of freedom and *P* value noted *Give P values as exact values whenever suitable.* |
| ☐ | ☒ | For Bayesian analysis, information on the choice of priors and Markov chain Monte Carlo settings |
| ☒ | ☐ | For hierarchical and complex designs, identification of the appropriate level for tests and full reporting of outcomes |
| ☒ | ☐ | Estimates of effect sizes (e.g. Cohen's *d*, Pearson's *r*), indicating how they were calculated |

*Our web collection on statistics for biologists contains articles on many of the points above.*

## Software and code

Policy information about availability of computer code

| Data collection | No software was used |
|---|---|
| Data analysis | Mammal species identification: Primer target-regions were trimmed from Sanger reads in Geneious Prime 2025.1.2, a similarity search performed in BLAST+ 2.16.0. |
| | Sequencing data analysis: Raw sequencing reads were quality-filtered using trimmomatic v0.39. Filtered reads were mapped to the most recent MPXV genome from TNP (GenBank accession number MN346702) using BWA MEM v0.7.17-r1188. Mapped reads were sorted and duplicates removed using SortSam and MarkDuplicates by Picard v2.13.3 (http://broadinstitute.github.io/picard/). In parallel, paired reads were mapped to the reference genome in Geneious Prime 2023.1.2 (https://www.geneious.com). |
| | The complete mitochondrial genome of the squirrel was de-novo assembled from quality-filtered reads using SPAdes v3.13.0. Oxford Nanopore reads were quality trimmed using BBDuk Trimmer v1.0 and de novo assembled using Flye v2.9.2. The entire dataset was then re-mapped against the initially generated sequence through Minimap2 v2.17. |
| | Diet analysis metabarcoding: For taxonomic assignement of sequences resulting from a metabarcoding experiment a custom pipeline using OBITools v3.0.1b2 was used (described in detail in the methods section of the paper). |
| | Phylogeny: Quality of MPXV sequences was evaluated through Nextclade v3.12.036 (https://clades.nextstrain.org/). Datasets were aligned using MAFFT v7.505n. We used Squirrel v1.2.2 (https://github.com/aineniamh/squirrel) to generate a masked alignment of 197211 positions. We used this alignment to reconstruct a maximum likelihood phylogeny using IQ-TREE v2.1.4b. Branch robustness was assessed by Shimodaira-Hasegawa-like approximate likelihood ratio tests. We ran a regression of root-to-tip distances vs. time and identified the best-fitting root of the resulting tree using TempEst v1.5.3. For molecular clock analyses were performed using Phylostems and BEAST v1.10.5 (described in detail in the methods section of the paper). We summarized the posterior set of trees under the form of a maximum clade credibility tree using TreeAnnotator v1.10.5 (distributed with BEAST). All trees were further visualized and edited in iTOL v7.1 (https://itol.embl.de/). |

Visualisation: Geographic maps, the molecular detection of MPXV in faeces, and sequencing coverage were plotted in RStudio v2022.02.3, ggplot2. Phylogenetic trees were visualised on iTOL v7.1 (https://itol.embl.de/). Figure design was modified in Inkscape v1.4. Photographic images were cut in Powerpoint 2016 and GIMP 2.10.38, videos were cut and captioned in Corel VideoStudio 2023.

For manuscripts utilizing custom algorithms or software that are central to the research but not yet described in published literature, software must be made available to editors and reviewers. We strongly encourage code deposition in a community repository (e.g. GitHub). See the Nature Portfolio guidelines for submitting code & software for further information.

## Data

Policy information about availability of data

All manuscripts must include a data availability statement. This statement should provide the following information, where applicable:
- Accession codes, unique identifiers, or web links for publicly available datasets
- A description of any restrictions on data availability
- For clinical datasets or third party data, please ensure that the statement adheres to our policy

Raw reads resulting from MPXV target enrichment in the samples, from the DNA metabarcoding, and from the shotgun sequencing of the viral isolates are available in the European Nucleotide Archive under project accession number PRJEB90150, sample accession numbers ERS24827514 - ERS24827784, ERS25063784 - ERS25063790, ERS25115682 - ERS25115684, and run accession numbers ERR15137312, ERR15137322 - ERR15137323, ERR15137325 - ERR15137330, ERR15137332 - ERR15137334, ERR15137343 - ERR15137588, ERR15172398 - ERR15172401,ERR15390135, and ERR15391827-ERR15391828. Consensus sequences of MPXV high quality genomes are available in Pathoplexus, accession numbers PP_0031VY3.1 (sooty mangabey faecal sample), PP_0031VZ1.1. (sooty mangabey necropsy) and PP_0031W1X.1 (fire-footed rope squirrel necropsy). Additional MPXV genomes included for phylogenetic analysis were retrieved from GenBank (accession numbers AY603973, AY741551, AY753185, DQ011153, DQ011156, KJ136820, KJ642614, KJ642615, KJ642616, KJ642617, KP849470, MG693723, MN346696, MN346699, MN346702, OP612690), and GISAID (accession numbers EPIISL19561127, EPIISL19561129, EPIISL19561136, EPIISL19630777, EPIISL19630778, EPIISL19729748 - EPIISL19729752). Maps were built using geographical data from OpenStreetMap, spatial data on squirrel species distribution was obtained from IUCN redlist and used in accordance with the IUCN Red List Terms of Use for non-commercial research (https://www.iucnredlist.org/terms/terms-of-use). Source Data for Fig. 3 are provided with the paper.

## Research involving human participants, their data, or biological material

Policy information about studies with human participants or human data. See also policy information about sex, gender (identity/presentation), and sexual orientation and race, ethnicity and racism.

| | |
|---|---|
| Reporting on sex and gender | This study did not include research involving human participants, their data, or biological material. |
| Reporting on race, ethnicity, or other socially relevant groupings | This study did not include research involving human participants, their data, or biological material. |
| Population characteristics | This study did not include research involving human participants, their data, or biological material. |
| Recruitment | This study did not include research involving human participants, their data, or biological material. |
| Ethics oversight | This study did not include research involving human participants, their data, or biological material. |

Note that full information on the approval of the study protocol must also be provided in the manuscript.

## Field-specific reporting

Please select the one below that is the best fit for your research. If you are not sure, read the appropriate sections before making your selection.

☐ Life sciences  ☐ Behavioural & social sciences  ☒ Ecological, evolutionary & environmental sciences

For a reference copy of the document with all sections, see nature.com/documents/nr-reporting-summary-flat.pdf

## Ecological, evolutionary & environmental sciences study design

All studies must disclose on these points even when the disclosure is negative.

| | |
|---|---|
| Study description | Here we describe an outbreak of MPXV in a group of wild sooty mangabeys (Cercocebus atys), in Taï National Park (Côte d'Ivoire), and the investigation of a potential source of this outbreak by looking for MPXV in sympatric rodents and shrews, and other wildlife found dead in Tai National Park. This work falls within the long term behavioural and health monitoring program of the Tai Chimpanzee Project and within a 2-year initiative aimed at sampling small terrestial mammals inside and around Tai National Park (BIODIV-AFREID). |
| Research sample | The research sample is represented by a group of wild sooty mangabeys (Cercocebus atys) of approximately 80 individuals, and of 694 rodents and shrews trapped within and around Tai National Park. We also sampled 23 carcasses found in Tai National Park. |
| Sampling strategy | We performed non-invasive sampling through collection of faeces from symptomatic and asymptomatic mangabeys. Necropsies on dead wildlife were performed by trained veterinarians. Rodents and shrews were trapped and sampled by biologists and |

veterinarians. Sampling was perfomed under anaesthesia with a minimally invasive technique which entailed collecting an oral and rectal swab, and blood. If the animals died in the trap, succumbed to anaesthetisation or were euthanised due to weakness, we performed a full necropsy.

| | |
|---|---|
| Data collection | Data collection was performed by local field assistants, researchers and veterinarians working for the Tai Chimpanzee project. Behavioral data were collected on apposite data sheets. Clinical data was collected by a veterinarian and documented via pictures and videos. Necropsy reports were written by the veterinarian who performed them. Data collection from the small mammal trappings was performed by biologists and veterinarians. |
| Timing and spatial scale | Sample collection for the mangabey habituation started in 2012 and has been routinely carried out ever since in the research area. Over 12 years we have accumulated a collection of mangabey faecal and urine samples. Necropsy samples from all wildlife found dead in the area have been collected since 1994, but this study only reports resuts of carcasses collected between 2019 and 2024 Rodent and shrew trapping was performed in Tai National Park, surrounding fields, and villages in July-November 2021, April and May 2022 and March to April 2023. |
| Data exclusions | No specifica data were excluded from the study. |
| Reproducibility | All real-time PCRs were performed at least in duplicate. In presence of doubtful results all tests were repeated from the first step. |
| Randomization | Randomization is not relevant for this type of study, which is based on investigating infectious causes of illness in wildlife. To maximize our chances of pathogen detection we sampled all individuals, whenever possible. |
| Blinding | Not applicable to this study. |

Did the study involve field work? ☒ Yes ☐ No

## Field work, collection and transport

| | |
|---|---|
| Field conditions | Tropical rainforest |
| Location | Tai National Park, Ivory Coast |
| Access & import/export | All research is conducted under the umbrella of a collaboration with Ivorian partners and health authorities. Samples are routinely exported to Germany for diagnostic purposes following international guidelines and prior official authorization through CITES permits, where necessary. |
| Disturbance | Activities conducted for this study were carried out as part of the Tai Chimpanzee Project. This project is aimed at collecting behavioural observations from wild magabeys through an habituation process which has been running since 2012. All samples and observations collected are done with the minimum disturbance to wildlife and the environment. Rodent and shrew trapping was also conducted in respect of international guidelines to minimize disturbance and stress for the animals and their enviroment. |

# Reporting for specific materials, systems and methods

We require information from authors about some types of materials, experimental systems and methods used in many studies. Here, indicate whether each material, system or method listed is relevant to your study. If you are not sure if a list item applies to your research, read the appropriate section before selecting a response.

## Materials & experimental systems

| n/a | Involved in the study |
|---|---|
| ☒ | Antibodies |
| ☒ | Eukaryotic cell lines |
| ☒ | Palaeontology and archaeology |
| ☐ | ☒ Animals and other organisms |
| ☒ | Clinical data |
| ☒ | Dual use research of concern |
| ☒ | Plants |

## Methods

| n/a | Involved in the study |
|---|---|
| ☒ | ChIP-seq |
| ☒ | Flow cytometry |
| ☒ | MRI-based neuroimaging |

## Animals and other research organisms

Policy information about studies involving animals; ARRIVE guidelines recommended for reporting animal research, and Sex and Gender in Research

| | |
|---|---|
| Laboratory animals | This study did not involve laboratory animals. |

| | |
|---|---|
| Wild animals | This wild mangabey group has been under human habituation since 2012. During the mpox outbreak, this group consisted of 80 individuals, including 15 infants, 22 juveniles, 13 subadults, and 30 adults. A team of field assistants and researchers is following the animals on a daily basis from a 7-meter distance, recording behavioural data and collecting faeces and urine samples whenever it is possible. In normal situations, each assistant or researcher has one or two focal individuals per day to collect data and samples from. In outbreak situations, monitoring efforts are reinforced and sampling is attempted from all symptomatic and asymptomatic individuals. Rodents and shrews (mainly adults and some juveniles and infants) living in Tai National Park and its surroundings were also sampled for this study. |
| Reporting on sex | Samples were collected from all individuals wherever possible, independent of sex. |
| Field-collected samples | Samples are collected upon defecation or urination of the mangabeys (at environmental temperature and during day time) and stored in 2 ml cryotubes. The research camps of the Taï Chimpanzee Project are equipped with liquid nitrogen tanks for storage of samples (around -200°C). Samples from the rodents and shrews are collected during day and night time and placed in 2 ml cryotubes and stored at room temperature (oral and rectal swabs in nucleic acid preserving buffer) or directly frozen in liquid nitrogen tanks (around -200°C) present at the field site (necropsy samples). Samples are then transported to Abidjan for temporary storage at the Centre Suisse de Recherches Scientifiques (-80°C) and subsequently shipped to Germany on dry ice (around -80°C) whenever someone is traveling. All experiments are still ongoing as they are part of longitudinal studies. |
| Ethics oversight | No ethical approval was required for this study since it is based on collecting observational data from wild mangabeys and minimally invasive samples from live wild rodents and shrews. The work was done in collaboration with local partners and under the permission of the Ivorian national park authority OIPR. The permission included a) mangabey sampling, which was performed non-invasively without disturbing the animals' natural behavior; b) small rodent trapping with minimally invasive sampling and sacrificing a small subset of rodents (5% of the trapped animals); and c) the collection of tissue samples from naturally dead animals found in the research area or from trapped animals that died in the traps or as a consequence of anaesthesia. Research permit numbers are reported in the acknowledgements section of the manuscript. |

Note that full information on the approval of the study protocol must also be provided in the manuscript.

## Plants

| | |
|---|---|
| Seed stocks | This study did not involve plants. |
| Novel plant genotypes | This study did not involve plants. |
| Authentication | This study did not involve plants. |

