## [Peer Review File · Nature]

Transmission of MPXV from fire-footed rope squirrels to sooty mangabeys

Corresponding Author: Professor Fabian Leendertz

Version 0:

Reviewer comments:

Referee #1

(Remarks to the Author)

This study synthesizes multiple, cumulative lines of evidence that collectively support a direct association between the rope squirrel reservoir and a monkeypox virus (MPXV) spillover event in non-human primates. The findings provide robust support for the identification of the MPXV reservoir more broadly, as well as for the identification of a specific outbreak's origin. Although the evidence may appear serendipitous, it is in fact the result of decades of rigorous cohort-based research. The study underscores the importance of sustained wildlife health monitoring and highlights the ongoing risk of zoonotic transmission at the human–animal interface, particularly in the context of the wild game trade. Furthermore, it contributes critical insight into the factors that facilitate spillover. We recommend that the manuscript be accepted pending major revisions, including a re-analysis of the phylogenetic component using more rigorous methods. We recommend Bayesian analyses as a necessity as it may strengthen the results and conclusions of the study, even though it could slow things down.

Major:

While the evidence to support the specific rope squirrel species as a reservoir host is strong, the authors should discuss the possibility that it is an intermediate host infected incidentally by an outbreak in another host in proximity, as this cannot necessarily be discounted – or discuss the evidence against this concern, which readers will certainly raise. This could be extended into a conversation about maintenance across different host species.

The study would be significantly strengthened by the inclusion of Bayesian phylogenetic analysis. This would allow the authors to estimate the timing of divergence between the viruses isolated from squirrels and non-human primates. Furthermore, a Markov jump analysis across the Bayesian posterior distribution could provide an estimate of the exact timing of the interspecies transmission event, with associated uncertainty. This analysis would enhance the interpretation of key aspects discussed in Lines 142–150, including the cryptic circulation and the temporal dynamics of symptom onset and outbreak progression.

There is no methodological justification for extracting only variable sites from the alignment for phylogenetic reconstruction. Doing so substantially reduces the phylogenetic signal, as conserved sites contribute essential information for branch length estimation, model parameterization etc. The analysis should be repeated using the full alignment. While the current inference is likely to hold, it should be supported by a more robust approach.

A clearer understanding of the mangabeys' meta-population structure and/or contact network, even if briefly outlined in the main text, would improve the reader's ability to interpret the transmission network discussed loosely and its temporal dynamics.

Lines 117–118, 134–135: To better assess whether the genomes are truly identical, more detailed information on genome coverage would be helpful—for example, the number of ambiguous sites. Including coverage plots in the extended data would add clarity, as the distribution of read depth provides more meaningful insight than an average coverage value alone. Figure 1: It would be helpful to show the distribution of other rope squirrel species, as there is limited evidence to suggest that only a single species within the genus is susceptible to MPXV infection. Including this information would provide a broader ecological context for interpreting the reservoir dynamics.

Lines 500–501: You mention using a read threshold of two for samples with shallow coverage. This is quite low and may introduce spurious bases (even though the genomes are identical – caveat, I'm not sure how many ambiguous sites there are). I would advise caution in relying on these sequences or making inferences about their similarity to others (whether identical or otherwise) without clearly acknowledging the low quality of these sequences. Additionally, please refer to the earlier comments regarding the need to address ambiguities and provide further clarification on the parameters used for consensus calling as in the minor comments below.

Minor:

The sentence from lines 59–62 could be rephrased for clarity, as the phrase "under the form" feels awkward.

Line 63-64: This has also been shown for Cameroon. It is worth citing: doi: 10.1101/2024.06.18.24309115

Line 72: I suggest you rephrase “famously”, as it is too colloquial.

Extended Data Fig. 3 seems to be a table, not a figure?

In Figure 3, changing the alpha for the filled data points would improve visibility, as the black color currently blocks the purple (e.g. third row from the bottom)

Line 113: “Reconstruct viral genomes” should be changed to “generate” or “assemble,” as it is not a reconstruction.

Line 191: It is worth highlighting these sequences in Figure 4.

Line 195: Remove “perhaps” for redundancy.

Line 499: The use of the term “base calling” is potentially misleading here, as it typically refers to the upstream process performed on the sequencing platforms. Are you referring to variant calling during consensus sequence generation? It would also be clearer to rephrase “minimum of 20 reads” as a “read depth threshold.” Additionally, the reference to “95% agreement” is unclear—does this refer to the nucleotide frequency threshold used for consensus calling? If so, please specify this explicitly. It would also be helpful to indicate the threshold used for assigning ambiguous nucleotides.

Referee #2

(Remarks to the Author)

I co-reviewed this manuscript with one of the reviewers who provided the listed reports.

Referee #3

(Remarks to the Author)

The authors describe the detection and sequencing of genetically identical Mpox virus (MPXV) in a single fire-footed rope squirrel found dead 12-weeks prior to the detection of an outbreak in sooty mangabeys in Taï National Park. The authors confirm consumption of fire-footed rope squirrels by mangabeys in this park via metabarcoding analysis of faecal samples. The authors detect both MPXV DNA and fire-footed squirrel DNA in two faecal samples from the suspected index case for the mangabey outbreak. As MPXV was shed in the feces of seven mangabeys with no observed clinical signs of mpox, it is unclear whether co-detection of MPXV DNA reflects the active infection status of the consumed squirrel and/or the mangabey. The authors found identical mpox virus sequences from samples collected 13 days apart in a mangabey necropsy sample (tissue not specified) and mangabey faecal sample in addition to the identical sequence from the fire-footed squirrel two weeks prior. It is unclear what the %coverage is for these sequences (we have average depth of coverage, but not completeness).

While I am enthusiastic about the ecological context including the fly sequencing, testing of mangabey faecal samples both for MPXV and diet, and virus isolation, I disagree with some conclusions in the paper including the downplaying of extensive previous work on MPXV ecology and a lack of formal conceptual framing. At present, this manuscript is more reflective of an interesting case study than a conclusive determination of the reservoir or reservoirs of MPXV. Rope squirrels have been implicated as reservoirs for both the clade I and clade II lineages of mpox virus, including detection by PCR in African rope squirrels over space and time in museum specimens (Tiee et al. 2018) and extensive serological surveillance data from wildlife sampling efforts. For me, the most convincing evidence for African rodents as potential maintenance hosts for clade II MPXV remains the introduction of MPXV to the United States through the exotic rodent trade in which naïve prairie dogs acquired MPXV through co-housing with infected African rodents.

I would like to see more tempered conclusions in this manuscript with a more nuanced discussion on how the authors define reservoir and other possible explanations for cross-species transmission of MPXV including multiple reservoir hosts, multiple possible spillover hosts, or even flies as has been suggested by this group in the past and likely motivated the collection and testing of maggots which are only briefly mentioned here. Additional surveillance in the fire-footed rope squirrel population, for instance, could shed light on whether the squirrels are capable of maintaining MPXV in the population or if they are also incidental “spillover” hosts. The authors underplay previous results linking *Funisciurus* sp. squirrels to MPXV maintenance over decades for samples collected from across West and Central Africa which would have provided important context for their findings and boosted the support for *Funisciurus* species as true reservoirs. I think this is a missed opportunity, particularly given that this study is reporting just two genomes and three isolates which have not been made available through a public biological reagent repository such as the European Virus Archive. Further, the authors draw a critical conclusion (“evidence of direct transmission”) from the detection of MPXV by PCR in a mangabey faecal sample that also contained mitochondrial DNA from a fire-footed rope squirrel, but do not provide sequence data from this key specimen.

I have a few additional, specific comments below:

On line 67, the authors state that “formal proof” is lacking to support rodents as a reservoir for mpox virus. It would be helpful to include a formal definition of “reservoir” and what evidentiary standards they require as “formal proof” for a reservoir. In my opinion, the epidemiological records are more convincing that imported African rodents initiated the 2003 mpox outbreak in the United States than the faecal sample with both fire-footed squirrel and MPXV DNA was the specific event that led to the outbreak in mangabeys in this study. As the authors point out, MPXV had been previously isolated from a Thomas’ rope squirrel in 1985 (not 1978 as written in text) 300 m from a village where a human mpox case had been confirmed years prior and PCR data confirm circulation of MPXV in *Funisciurus* squirrels in Central Africa.

Line 175: If the data exist, they should be included in this publication. Unpublished data are not acceptable evidence to

support conclusions

Line 183: personal communications are not acceptable evidence, please provide direct observation or cite from existing published literature to support this claim.

Lines 437 & 439: please include the primer/probe sequences in text so the work can be replicated without encountering a second paywall – particularly important for work relevant for and conducted in the Global South

The methods section lacks sufficient detail in general.

Supplemental table 4 is inadequate, please include basic taxonomic identification for each individual sampled; even if species-level identification is not possible the researchers should include highest resolution possible (e.g. genus or family) rather than a blank space.

Referee #4

(Remarks to the Author)

General comments

This study describes a remarkable investigation into the origin of a mpox outbreak in wild monkeys in West Africa. The diversity and quantity of laboratory methods directed towards the outbreak is impressive, going well beyond the norm for studies of wildlife diseases. Through longitudinal sampling, the authors stitch together a timeline and argue that the outbreak started when a monkey in the group ate an infected fire-footed rope squirrel. As the authors note, it is rare to have such temporal resolution in any wildlife disease outbreak investigation.

The paper is well structured and well written, and the laboratory analyses are sophisticated and compelling. The figures are clear and interesting (especially the photos of animals in the forest), and colour is used well. The study will certainly rank among one of the most comprehensive wildlife disease outbreak investigations on record.

A few overarching issues detract from the manuscript. First, in many sections the text reads as self-serving and self-congratulatory. The authors repeatedly stress that their foresight and superior research program in Ivory Coast is the reason why they were able to conduct this study, and that they are unusual in this respect. The effect is to be off-putting and to sound a bit “preachy” (e.g. lines 204-207). The data speak for themselves, so I suggest that the authors remove sections that tout their own preparedness and achievements. The value of long-term wildlife monitoring would make a good topic for a review paper or an opinion piece, but it is distracting here.

More centrally, perhaps, is the conclusion that fire footed rope squirrels are a MPXV reservoir, which is the main finding of the study, as evidenced by the title. As interesting and compelling as the data are, they don't support this conclusion. In traditional terms, a reservoir is a population that naturally harbors a pathogen at high enough prevalence that it can serve as a source of infection for other species (“spillover”). Usually, the reservoir species carries the pathogen without clinical consequence, even though the pathogen can be deadly when transmitted to other species (e.g. Marburg virus in bats). In this study, the authors have documented virus infection in a single fire-footed rope squirrel that was found dead. The cause of death was unclear, but it is suspicious that MPXV might have killed it, in which case this goes against the idea that reservoirs typically aren't harmed by the pathogen for which they serve as a reservoir. Of course, there is the “multispecies reservoir concept” as well, in which a pathogen is maintained through transmission within and among several species (to which the authors allude in lines 187-189 but do not discuss further, as they perhaps should have). In that case, the fire footed rope squirrel may simply have been one of many reservoir species involved. Finally, the data are also consistent with the fire footed rope squirrel not being a reservoir at all. Perhaps there was an outbreak originating from another reservoir, and the fire footed rope squirrel 3 km south of the territory was a victim, just like the monkeys. Even if the monkey outbreak began with consumption of a fire-footed rope squirrel, maybe the monkey that ate the squirrel found it dead already, and the fire-footed rope squirrel had become infected from another, still unknown reservoir. To claim that the fire footed rope squirrel is a reservoir would require examining a large number of fire footed rope squirrels and a large number of other potential reservoirs in the forest. This study does not have such data, such that concluding that the fire footed rope squirrel is a reservoir is premature.

Another point related to the above: how confidently can the 16s sequence obtained from diet analysis be attributed to the fire-footed rope squirrel? From lines 112-128 of the supplementary “diet analysis” section, it appears that the inference is based on a 91 bp segment of the 16s gene (which does BLAST closest to an unidentified squirrel from DRC). Could this segment be identical in multiple species of squirrels or even other rodents in the forest? As the authors' BLAST analysis shows, online sequence databases are incomplete. It is therefore possible that this sequence is a perfect match to sequences of several local sciurids, including but not limited to the fire-footed rope squirrel.

Finally, I'm not very familiar with the hybridization capture method described, but could it create biases in the MPXV genome sequences that make them seem identical when they are, in fact, not? Looking at how the method works, it appears that it cannot detect gene duplications or rearrangements and that it may preferentially enrich for conserved viral genome regions. If the identical nature of the MPXV genomes might be due to such biases, the authors should discuss this.

Directed comments

Lines 48-50. Consider deleting this sentence because it is a bit self-congratulatory and does not add information.

Lines 52-53. The second phrase of this sentence is a bit too strongly worded, and the conclusion it makes strays too far from the data. Consider changing this phrase to a statement about risk, such as “exposure to these animals likely represents a risk for zoonotic transmission of mpox.”

Line 61. “Potent” may not be the best word choice here. Consider “useful,” or simply delete.

Line 62. It’s not clear to what “two pillars” refers. Change wording to clarify.

Line 68. The role of African rodents as MPXV reservoirs is more than an assumption. There is decent evidence to support this, beyond the two papers the authors cite. Rather than “formal proof to support this assumption is lacking,” consider something more nuanced, such as “direct evidence from field studies has been rare.”

Lines 69-71. This sentence is too hyperbolic. Depending on what the reservoir(s) is/are, and what the sociopolitical context is, knowing the reservoir might not prevent MPXV transmission or thwart outbreaks. Consider re-writing in a more nuanced way, or delete.

Line 100. Which protocols were these, and is there a citation?

Line 108. Change “babies” to “infants,” to be consistent.

Lines 187-189. Multispecies model. May be no single reservoir.

Figure 3. It looks like an adult animal (sixth row of the “Adults” section of the figure) tested positive for MPXV before the “index case” in red. This would seem to contradict the statement in lines 161-163. It’s difficult to try to figure this out from the supplemental tables.

Version 1:

Reviewer comments:

Referee #1

(Remarks to the Author)

We still have major concerns regarding the phylogenetic analyses that would benefit from further clarification and re-analyses.

Line 611: “Sites comprising ambiguities or gaps were removed in Geneious, resulting in an alignment of 134477 positions.” You have removed approximately 70 000 sites from the alignment. Were all sites containing ANY ambiguities or gaps in any sequence removed? The sites removed are substantially more than just the terminal repeats, which is the convention to mask (~7kb). What is the methodological justification for this? Missing information or gaps in columns will simply not contribute phylogenetic information for those sequences, while site patterns in other sequences add information and resolution. For terminal repeats and areas of low complexity, the correct approach is to mask your alignment. You can try <https://github.com/aineniameh/squirrel>. Again, the inference will likely hold, but there is no methodological justification for performing phylogenetics in this manner, especially as most of the inference is regarding divergence aka branch lengths. You cannot say sequences are identical, if you’re removing parts of the alignment without justification?

All of the results, including the phylodynamic analyses, will be affected by this, particularly those addressing the “near-identicalness” of the genomes, and this issue should be directly addressed. I should also note that the consensus frequency threshold of 0.95 is quite conservative and may have excluded sites with lower read depth or true mixed populations, which are not necessarily indicative of contamination.

Line 626: Was the molecular clock unidentifiable or did it fail to converge? In either case, this is likely due to the exclusion of much of the phylogenetic backbone and therefore signal by focusing on just 23 sequences with low diversity. What was the justification for excluding other sequences from the analysis, particularly given that there was an overall positive temporal signal? I suspect that the rate heterogeneity in the reservoir host might not be strong enough in the shorter term in this subset to necessitate a relaxed clock, but rate heterogeneity should still be tested for. You can normally get an initial feel for it from the temporal regression – which should be included in the supplementary for evaluation as well. Additionally, the sampling time difference between the “near-identical” sequences should also be discussed in this context, to give the reader an expectation of divergence along the timescale given the evolutionary rate.

We believe the expanded discussion on alternative hosts, including other squirrel species, and the multi-species maintenance ecology sufficiently addresses our concerns in that area.

Referee #3

(Remarks to the Author)

The authors have sufficiently addressed my concerns and I congratulate them on this excellent and exciting work.

Referee #4

(Remarks to the Author)

The authors have done an excellent job responding to critiques by providing new analyses and providing new explanatory text. The manuscript is extremely thorough and an impressive piece of work.

However, despite their rebuttal, the authors still are not justified in saying that they have shown the fire-footed rope squirrel to be a reservoir. Their responses to items 28 and 29 in the rebuttal only reiterate the remarkable story of sylvatic transmission in this outbreak; they do not strengthen the reservoir argument. There is little doubt that there is a connection between fire footed rope squirrels and this primate outbreak. However, that connection need not be explained by the fire footed rope squirrel being a reservoir. The fire footed rope squirrel might be a reservoir, but the data here (as impressive as they are) simply don't show that. The authors have added a definition of reservoir in lines 72-74, which is a good addition. That definition states that a criterion is "the virus can circulate permanently" in the host. The authors have not provided evidence of permanent (sustained) circulation of MPXV in the fire-footed rope squirrel. They have presented strong circumstantial evidence that a single fire footed rope squirrel nearby was infected with MPXV prior to the monkey outbreak. Furthermore, the study of museum specimens that the authors cite is not about fire footed rope squirrels. It is about multiple species of rope squirrels, of which the fire footed is but one. In that study, 8/201 fire footed rope squirrels collected over 120 years across Central Africa tested positive for MPV. This shows a low period prevalence, and not "permanent circulation." As strong as the clues are that fire footed rope squirrels were involved in this outbreak, the conclusion that the fire footed rope squirrel is a reservoir is not supported by the data.

Along these same lines, the original title of the manuscript is too strong. The authors should not say "are a reservoir." The proposed alternative title is also too strong. As impressive as the data are, they are circumstantial and do not demonstrate direct causality. The authors should change the title to something more objective, such as "Monkeypox virus outbreak in wild primates linked to transmission from fire-footed rope squirrels" (example only). It's clear that the authors want to state that they have identified the first MPXV reservoir (see lines 74-75, "According to this definition, no MPXV reservoir has yet been identified"). However, as remarkable as the data are, the study still falls short of being able to attribute reservoir status to the fire-footed rope squirrel.

Please remove all statements in the title and elsewhere that go beyond the data presented in implicating the fire footed rope squirrel as a reservoir. The paper will be stronger for it.

Version 2:

Reviewer comments:

Referee #1

(Remarks to the Author)

I have reviewed the authors' responses to my last round of queries and comments and found them satisfactory.

Dear Editor,

We thank you for your and the reviewers' feedback on the initial version of our manuscript, and for the offer to revise it.

We have done our best to address the major criticisms raised in the first round of reviewing. Specifically, we have added Bayesian phylogenomic analyses, and revised the text to present a more nuanced interpretation of earlier findings, as well as of our own.

We have also clarified and amended a number of other minor points mentioned by the reviewers.

You will find below a point-by-point answer to the reviewers' comments. To allow an easier reading of this document, we have **colored in blue the Reviewers' and Editor's comments**, and replied in black. For each comment, where relevant, we provide references to the lines of the new version of the manuscript where changes can be found. As suggested by the Editor, we provide two versions of the manuscript: one where changes are tracked or **highlighted**, and one clean version. Line numbers mentioned in our replies refer to the version of the manuscript where changes are tracked or highlighted. To facilitate identifying the modifications in the manuscript, we have numbered them according to the corresponding comment number assigned in this document.

We think that the paper has gained in strength, clarity and nuance. We hope that you will share this feeling, and deem it worthy of publication.

With our best regards,

Fabian H. Leendertz and Livia V. Patrono (on behalf of all coauthors)

Dear Professor Leendertz

Your manuscript, "Fire-footed rope squirrels (*Funisciurus pyrropus*) are a reservoir host of monkeypox virus (*Orthopoxvirus monkeypox*)", has now been seen by 4 referees, whose comments are attached below. While they find your work of potential interest, as do we, they have raised important concerns that in our view need to be addressed before we can consider publication in Nature.

The reviewers find the paper of interest, but they also flag inadequate acknowledgement of prior literature documenting MPXV infection in fire-footed rope squirrels as well as lack of definitive evidence that the squirrels are indeed a reservoir for the virus.

We have taken into consideration the comments of the reviewers and have significantly re-written parts of the manuscript to (1) acknowledge previous literature and (2) clarify important definitions such as that of a reservoir.

The reviewers also feel that the work could be improved with additional analysis, particularly regarding the genomic analysis.

We have now run additional analyses, which we present in the main text and as an additional Extended Data Figure 8. We also provide more complete information on the genomes we assembled from the mangabey and squirrel samples, including a detailed description of genome comparisons. This information is presented in the main text, as Extended Data Figures 5 and 6, and in the Supplementary

Information file. To address the reviewers' concerns on the identification of the fire-footed rope squirrel DNA in the faeces of the mangabeys, we now provide an alignment including the sequences generated in our study and those of related squirrel species present in the country, showing that the DNA we detected is unambiguously that of a fire-footed rope squirrel (see replies to reviewers' comments and the supplementary information file).

Should you be willing to do the additional analysis as well as presenting the work more carefully in line with the reviewers' comments we would be interested in considering a revised manuscript (unless something similar has been accepted at Nature or appeared elsewhere in the meantime). We hope to receive your revised paper within four to six months. If you cannot complete the required revisions within this time frame, please let us know when you would anticipate being able to submit a revised manuscript.

We now provide the additional analyses requested by the reviewers, and a revised version of the text that builds on the suggestions of the reviewers. We suggest keeping the original title, but if the Editor and Reviewers find it more appropriate we would propose the following alternative title: Fire-footed rope squirrels (*Funisciurus pyrropus*) cause a monkeypox virus (*Orthopoxvirus monkeypox*) outbreak in wild sooty mangabeys (*Cercocebus atys*)

Point-by-point answer

Referee #1

1. This study synthesizes multiple, cumulative lines of evidence that collectively support a direct association between the rope squirrel reservoir and a monkeypox virus (MPXV) spillover event in non-human primates. The findings provide robust support for the identification of the MPXV reservoir more broadly, as well as for the identification of a specific outbreak's origin. Although the evidence may appear serendipitous, it is in fact the result of decades of rigorous cohort-based research. The study underscores the importance of sustained wildlife health monitoring and highlights the ongoing risk of zoonotic transmission at the human–animal interface, particularly in the context of the wild game trade. Furthermore, it contributes critical insight into the factors that facilitate spillover. We recommend that the manuscript be accepted pending major revisions, including a re-analysis of the phylogenetic component using more rigorous methods. We recommend Bayesian analyses as a necessity as it may strengthen the results and conclusions of the study, even though it could slow things down.

We thank the referee for their positive words and careful reading of the manuscript.

2. Major: While the evidence to support the specific rope squirrel species as a reservoir host is strong, the authors should discuss the possibility that it is an intermediate host infected incidentally by an outbreak in another host in proximity, as this cannot necessarily be discounted – or discuss the evidence against this concern, which readers will certainly raise. This could be extended into a conversation about maintenance across different host species.

We thank the reviewer for their suggestion, which we agree should be addressed in the text. We have now extensively re-written the first part of main text to bring in previous evidence of MPXV circulation in distinct African rodent species, including serological, PCR, and ecological niche modeling data (lines 83-92). We now also explicitly mention in the text that we cannot rule out that the MPXV-positive squirrel may have acquired the infection from another host (lines 181-185). We finally bring up again the idea of a multi-host maintenance, which should be further investigated, in lines 256-258.

3. The study would be significantly strengthened by the inclusion of Bayesian phylogenetic analysis. This would allow the authors to estimate the timing of divergence between the viruses isolated from squirrels and non-human primates. Furthermore, a Markov jump analysis across the Bayesian posterior distribution could provide an estimate of the exact timing of the interspecies transmission event, with associated uncertainty. This analysis would enhance the interpretation of key aspects discussed in Lines 142–150, including the cryptic circulation and the temporal dynamics of symptom onset and outbreak progression. There is no methodological justification for extracting only variable sites from the alignment for phylogenetic reconstruction. Doing so substantially reduces the phylogenetic signal, as conserved sites contribute essential information for branch length estimation, model parameterization etc. The analysis should be repeated using the full alignment. While the current inference is likely to hold, it should be supported by a more robust approach.

We have welcomed the suggestions of the reviewer and have run an additional set of analyses to reinforce our study. We acknowledge that running the analyses without formally accounting for the proportion of identical sites removed may influence the phylogenetic signal. First, we re-ran ML analyses without removing identical sites from the alignment, and used the resulting tree for the new Figure 4, which

essentially reinforces the information previously obtained. We then ran Bayesian analyses using molecular clocks to estimate the tMRCA of the squirrel- and mangabey-infecting MPXV. The results are now presented in the main text (lines 191-196) and as Extended Data Fig. 8.

We agree that Markov jumps and rewards are in general a great tool to summarize the time spent in states (here, hosts) in a phylogeny or even individual branches. However, we think that in this specific case this method may not be appropriate.

Indeed, since the squirrel- and sooty mangabey-derived MPXV sequences are nearly identical (please see our reply to comment #5 and the genomic characterization paragraph in the supplementary information), one can assume that MPXV being such a slowly evolving virus their most recent common ancestor also had the same exact sequence. If it didn't, either 1) both the squirrel- and sooty mangabey-infecting viruses should have mutated at the same exact spot in the genome – a highly unlikely scenario, or 2) one of or both viruses should have experienced two mutations (an initial mutation followed by a reversal) – also a highly unlikely scenario.

Accordingly, the most likely scenario is that no mutation happened between the MRCA of these sequences and the sequences themselves. It is our understanding that this means there is no information available to estimate when the host transition happened on the MRCA to squirrel branch (it is an actual zero length-branch), i.e. no information to estimate a proper Markov reward for this host transition.

Therefore, we believe that we cannot better resolve the timing of this host jump on a purely genomic basis than by providing a lower estimate (0 year – it happened at the very end of the branch leading to the squirrel-infecting virus) and an upper estimate represented by the tMRCA (median 2021).

Of course, the other additional lines of evidence that we provide complete the genomic picture and lead us to conclude that we have caught MPXV transmission in the act – and therefore that the host jump date is known with the very high precision suggested by the fecal analyses' results (a few days, and perhaps even a single day).

4. A clearer understanding of the mangabeys' meta-population structure and/or contact network, even if briefly outlined in the main text, would improve the reader's ability to interpret the transmission network discussed loosely and its temporal dynamics.

We agree that this information would be useful for the reader to understand the context of the outbreak, and have added a description of their social system to the manuscript (lines 118-119). In contrast to the fission fusion dynamics that characterize chimpanzee communities, sooty mangabeys live in stable, large multi-male multi-female social groups and females typically remain in their natal groups for life, while males disperse away from their social groups before reaching sexual maturity. Females form linear, despotic, stable matrilineal hierarchies. We agree that aspects of their social behavior would likely have influenced transmission within the group (e.g., grooming networks, contact networks, sexual networks), but understanding the transmission of the virus within the group after transmission occurred was not the focus of the current paper and we have not (yet) attempted to analyze the behavioral data available from before/during/after the outbreak, while still giving the reader useful context.

5. Lines 117–118, 134–135: To better assess whether the genomes are truly identical, more detailed information on genome coverage would be helpful—for example, the number of ambiguous sites.

Including coverage plots in the extended data would add clarity, as the distribution of read depth provides more meaningful insight than an average coverage value alone.

We thank the reviewer for suggesting to add a more detailed description of the genomes assembled from the mangabeys and the squirrel. We have now added more comprehensive information on genome coverage in the main text (lines 153-160 and 186-188), and a supplementary discussion point to present detailed genome comparisons (see Supplementary Information file). We have also generated coverage plots for each sample, which we now present as Extended Data Fig. 5 and 6. In the review of the assemblies, we identified four positions in the MPXV genome where there are small differences between the MPXV found in the mangabeys and the one found in the fire-footed rope squirrel. The identified differences are all small indels present in regions characterized by nucleotide tandem repeats (see the newly added section on genomic comparison in the Supplementary Information for details), which are notoriously known to pose challenges to MPXV genome assemblies. During the review process, we checked thoroughly all these regions and edited the consensus sequences (some of which contained stretches of Ns due to a read misalignment) to reflect the information present in the reads. In some instances, we found mixed populations of reads with different tandem repeat numbers, which we also report in the supplementary genomic description. We note here that regions with tandem repeats are typically excluded from alignments used for MPXV phylogenetic analyses (and were in ours), and that indels are, in fact, not informative in this framework. We also observe that the identification of two nearly identical but distinct variants (one found in a dead fire-footed rope squirrel, the second in sooty mangabeys exposed through diet to the same squirrel species) reinforces the hypothesis of fire-footed rope squirrels being natural host harboring diverse MPVXs.

6. Figure 1: It would be helpful to show the distribution of other rope squirrel species, as there is limited evidence to suggest that only a single species within the genus is susceptible to MPXV infection. Including this information would provide a broader ecological context for interpreting the reservoir dynamics.

We thank the reviewer for this suggestion. While we agree that it is interesting to provide a broader ecological context for potential MPXV circulation in rope squirrels, including the distribution of all *Funisciurus* sp. in Fig.1 would make this figure rather crowded (we indeed tried to plot all the species on the map). We have now added an Extended Data Figure (#1 of the revised manuscript) with separate maps showing the distribution of other rope squirrel species.

7. Lines 500–501: You mention using a read threshold of two for samples with shallow coverage. This is quite low and may introduce spurious bases (even though the genomes are identical – caveat, I’m not sure how many ambiguous sites there are). I would advise caution in relying on these sequences or making inferences about their similarity to others (whether identical or otherwise) without clearly acknowledging the low quality of these sequences. Additionally, please refer to the earlier comments regarding the need to address ambiguities and provide further clarification on the parameters used for consensus calling as in the minor comments below.

We understand and share the concerns of the reviewer regarding the use of lower quality genomes. While we did generate partial viral genomic information from some of the mangabey samples and report them as part of the outbreak investigation, the phylogenomic analyses were performed only on high quality genomes. We apologize for not providing a clearer overview of the generated data in the first version of the manuscript. We have now added more details on genome coverage for each sample, and the criteria

used for consensus calling throughout the text and methods. We have also added a supplementary paragraph with a detailed description of the genomes generated from mangabey and squirrel samples, and their comparison. As visual aid to the text, we now provide coverage plots as Extended Data Fig. 5 and 6.

Minor:

8. The sentence from lines 59–62 could be rephrased for clarity, as the phrase "under the form" feels awkward.

We have substituted “under the form” with “identifiable as” in line 63.

9. Line 63-64: This has also been shown for Cameroon. It is worth citing: doi: 10.1101/2024.06.18.24309115

We thank the reviewer for suggesting this preprint, which we have now added to the main text (line 67).

10. Line 72: I suggest you rephrase “famously”, as it is too colloquial.

We have removed “famously” from the sentence, which is now found in lines 107.

11. Extended Data Fig. 3 seems to be a table, not a figure?

Indeed, Extended Data Fig. 3 is a display item in table format. Since the information displayed is a summary of the faecal samples tested for MPXV DNA presence, we believe this format would be more appropriate for rapid visualization of the total numbers while reading the manuscript rather than having to open a separate excel file. We now provide also an excel table with the same content for the reviewer’s consideration. In case they thought it would be more suitable, we can remove the extended data figure and just refer to a supplementary table.

12. In Figure 3, changing the alpha for the filled data points would improve visibility, as the black color currently blocks the purple (e.g. third row from the bottom)

We thank the reviewer for their suggestion, which led us to modify this figure to facilitate the readers’ understanding of the data presented. In the new version of Fig. 3, we have simplified the color scheme and chose a light grey (instead of black) to plot the negative samples.

13. Line 113: "Reconstruct viral genomes" should be changed to "generate" or "assemble," as it is not a reconstruction.

We have now changed the “reconstruct” with “assemble”, line 149.

14. Line 191: It is worth highlighting these sequences in Figure 4.

These sequences are now marked with an asterisk in the new ML tree provided (Figure 4).

15. Line 195: Remove “perhaps” for redundancy.

We have removed “perhaps” from the sentence (see line 264).

16. Line 499: The use of the term “base calling” is potentially misleading here, as it typically refers to the upstream process performed on the sequencing platforms. Are you referring to variant calling during consensus sequence generation? It would also be clearer to rephrase “minimum of 20 reads” as a “read depth threshold.” Additionally, the reference to “95% agreement” is unclear—does this refer to the nucleotide frequency threshold used for consensus calling? If so, please specify this explicitly. It would also be helpful to indicate the threshold used for assigning ambiguous nucleotides.

We apologize that our consensus calling criteria were not clearly expressed. We have reviewed the text to specify that our criteria for consensus calling were set to a minimum unique read depth threshold of 20, and a nucleotide frequency of at least 95% (meaning that for a given nucleotide to appear in the consensus it would have to be present in at least 95% of the reads). We have also clarified that if a nucleotide at any given position would be present in <95% of the reads, an ambiguous base would automatically be inserted.

Referee #2 (Remarks to the Author):

I co-reviewed this manuscript with one of the reviewers who provided the listed reports.

We thank the reviewer for their comments and suggestions to improve our manuscript.

Referee #3 (Remarks to the Author):

17. The authors describe the detection and sequencing of genetically identical Mpox virus (MPXV) in a single fire-footed rope squirrel found dead 12-weeks prior to the detection of an outbreak in sooty mangabeys in Tai National Park. The authors confirm consumption of fire-footed rope squirrels by mangabeys in this park via metabarcoding analysis of faecal samples. The authors detect both MPXV DNA and fire-footed squirrel DNA in two faecal samples from the suspected index case for the mangabey outbreak. As MPXV was shed in the feces of seven mangabeys with no observed clinical signs of mpox, it is unclear whether co-detection of MPXV DNA reflects the active infection status of the consumed squirrel and/or the mangabey.

Given the very low viral copy number in that first sample (where we co-detected fire-footed rope squirrel DNA), followed by rising viral copy numbers in the faecal samples collected from the same mangabey in the following days (reported in Supplementary table 1), we hypothesize that the very first viral detection reflects the consumption of a MPXV-infected fire-footed rope squirrel, later followed by viral replication in the mangabey. Regarding MPXV shedding in faeces in the absence of clinical signs, this is something we have observed before in MPXV outbreaks in sympatric wild chimpanzees. Due to the observational nature of our study, we cannot say for each individual mangabey where we detect DNA shedding in faeces, whether this reflects an active infection or just transit of viral particles through the gut as a consequence of grooming (we observed many individuals eating crusts or pus from other group members). However, we believe this remains a clear indication of exposure in the group, which is supported by the detection of MPXV in three dead mangabeys.

18. The authors found identical mpox virus sequences from samples collected 13 days apart in a mangabey necropsy sample (tissue not specified) and mangabey faecal sample in addition to the identical sequence from the fire-footed squirrel two weeks prior. It is unclear what the % coverage is for these sequences (we have average depth of coverage, but not completeness).

We apologize for providing incomplete information on the genomes generated. We have now specified in the text that the necropsy sample from which we obtained the nearly complete genome is a skin sample (line 152). The detailed information on which samples were used to generate genomic information is also available in the supplementary excel tables 1, 2 and 5. We have also added information on the % of the genome covered with respect to a reference genome, and wrote an additional paragraph in the supplementary information describing the generated genomes and their comparisons. As visual aid to the text, we also generated coverage plots that are now provided as Extended Data figure 5 and 6.

19. While I am enthusiastic about the ecological context including the fly sequencing, testing of mangabey faecal samples both for MPXV and diet, and virus isolation, I disagree with some conclusions in the paper including the downplaying of extensive previous work on MPXV ecology and a lack of formal conceptual framing. At present, this manuscript is more reflective of an interesting case study than a conclusive determination of the reservoir or reservoirs of MPXV. Rope squirrels have been implicated as reservoirs for both the clade I and clade II lineages of mpox virus, including detection by PCR in African rope squirrels over space and time in museum specimens (Tee et al. 2018) and extensive serological surveillance data from wildlife sampling efforts. For me, the most convincing evidence for African rodents as potential maintenance hosts for clade II MPXV remains the introduction of MPXV to the United States through the exotic rodent trade in which naïve prairie dogs acquired MPXV through co-housing with infected African rodents.

We thank the reviewer for the appreciation of our study and for their suggestions, which we have taken into account to significantly re-write the first part of the manuscript. We acknowledge having only briefly touched on previous evidence resulting from other MPXV ecology studies, and adding more detailed information from these studies can indeed give the readers the necessary context to understand the data we present and discuss. In the revised version of the manuscript, we describe previous evidence supporting the hypothesis of rope squirrels are natural hosts of MPXV (lines 83-92), and include a detailed description of the 2003 USA outbreak investigation (lines 93-106). We concur that the latter is indeed the only well-characterized example of MPXV emergence in humans. However, it happened outside of the areas where MPXV is endemic, and under what can only be understood as extremely unusual conditions since the immediate reservoir for humans was prairie dogs. In addition, the reservoir for prairie dogs could not be narrowed down to a single rodent species. Taken together, we do agree that our initial presentation failed to stress sufficiently how much was already suspected on plausible grounds, but we are also convinced that our study now presents a compelling, until now missing, demonstration of the notion brought forth by our predecessors. As suggested by reviewers #1 and 2, we now also provide an additional map showing the range of the different rope squirrel species (Extended Data Fig. 1).

20. I would like to see more tempered conclusions in this manuscript with a more nuanced discussion on how the authors define reservoir and other possible explanations for cross-species transmission of MPXV including multiple reservoir hosts, multiple possible spillover hosts, or even flies as has been suggested by this group in the past and likely motivated the collection and testing of maggots which are only briefly mentioned here. Additional surveillance in the fire-footed rope squirrel population, for instance, could shed light on whether the squirrels are capable of maintaining MPXV in the population or if they are also incidental “spillover” hosts. The authors underplay previous results linking *Funisciurus* sp. squirrels to

MPXV maintenance over decades for samples collected from across West and Central Africa which would have provided important context for their findings and boosted the support for *Funisciurus* species as true reservoirs.

We thank the reviewer for their insights and suggestions on how to improve the manuscript, which we have integrated to build a richer and more nuanced structure of the text. As mentioned in the reply to comment #19, we have significantly re-written the introductory part of the manuscript to provide a better overview of previous literature and of the ecological context, which points at multiple species of African rodents, and especially rope squirrels, being involved in MPXV sylvatic maintenance. We have further modified the text to bring in a definition of reservoir, which following Haydon and colleagues, we define as a natural host in which the virus can circulate permanently and from which transmission to another host - humans in the case of zoonoses - is possible and documented (lines 78-80). We also state that even if indeed we cannot exclude that the squirrel we found dead may have acquired MPXV infection from another host, the overall accumulated evidence from this and other studies supports the hypothesis of rope squirrels being natural hosts of MPXV (lines 182-185).

We agree that further studies are required to understand MPXV maintenance in fire-footed rope squirrel populations, including its pathogenicity, and that broader search for MPXV in other small mammal hosts should continue, since the virus is very likely maintained in several species (see lines 256-258). While we share the excitement of the reviewer on the potential role of flies, we believe the evidence we have is not sufficient to put forward concrete hypotheses on their contribution to transmission or viral movement in the forest, and therefore prefer not to include sentences on this topic that would for now, remain rather speculative.

21. I think this is a missed opportunity, particularly given that this study is reporting just two genomes and three isolates which have not been made available through a public biological reagent repository such as the European Virus Archive. Further, the authors draw a critical conclusion (“evidence of direct transmission”) from the detection of MPXV by PCR in a mangabey faecal sample that also contained mitochondrial DNA from a fire-footed rope squirrel, but do not provide sequence data from this key specimen.

We apologize for the delay in submitting the data to public repositories. We have made the data available in the European Nucleotide Archive (ENA) and Pathoplexus. Please see the Data Availability statement in the manuscript for specific references. At the time of submission of the revised manuscript (July 24th, 2025), the sequencing data generated with the PromethION has not yet been released because of issues during the submission to ENA, which we are confident to resolve within a few days. Missing accession numbers will be provided as soon as received. Regarding the faecal sample in which we detected both MPXV and fire-footed rope squirrel DNA, we now provide in the supplementary information of the manuscript the following alignment showing the sequences of the targeted 16S region from the fire-footed rope squirrel carcass, from the sooty mangabey faecal sample (named metabarcoding sequence), and from the most closely related sciurid sequences. Since reviewer #4 raised the question of whether the targeted 16S sequence could be identical in other squirrel species, we looked into 16S sequences of local sciurids. There are only 9 sciurid species known to exist in Côte d’Ivoire (IUCN Red List data), and the fire-footed rope squirrel is the only representative of the *Funisciurus* genus. We could retrieve 16S sequences from 5 of the 6 genera present in the country (the missing genus, *Epixurus*, is not the closest genus to *Funisciurus*). This confirmed that the sequence we obtained from the mangabey faecal sample is a 100% match only to the fire-footed rope squirrel we determined from the dead animal found in the same forest. The next closest sequence was also generated from a fire-footed rope squirrel (of unknown provenance),

and already differed at 10 positions, and other species sequences of course showed even greater divergence.

I have a few additional, specific comments below:

22. On line 67, the authors state that “formal proof” is lacking to support rodents as a reservoir for mpox virus. It would be helpful to include a formal definition of “reservoir” and what evidentiary standards they require as “formal proof” for a reservoir. In my opinion, the epidemiological records are more convincing that imported African rodents initiated the 2003 mpox outbreak in the United States than the faecal sample with both fire-footed squirrel and MPXV DNA was the specific event that led to the outbreak in mangabeys in this study. As the authors point out, MPXV had been previously isolated from a Thomas’ rope squirrel in 1985 (not 1978 as written in text) 300 m from a village where a human mpox case had been confirmed years prior and PCR data confirm circulation of MPXV in Funisciurus squirrels in Central Africa.

We thank the reviewer for raising these relevant points, and acknowledge that these are important definitions that should be provided. We have now included a definition of “reservoir” at the beginning of the manuscript (please see lines 78-80 and our reply to comment #20). We agree with the reviewer that the investigations of the 2003 USA outbreak generated very important data on the role of African rodents as a source of MPXV, and have added an entire paragraph on this in the revised version of the manuscript (lines 93-106). However, these investigations did not provide evidence of a direct transmission of the virus from the African rodents to humans, since all human cases were caused by contact with infected prairie dogs, and we do not know which of the three species (dormice, giant pouched rat or rope squirrels) infected the latter, and potentially the other two rodent species. In the revised manuscript, we present all suggested evidence and hope the reviewer will appreciate these changes. We apologize for the wrong date of the isolation of MPXV for a Thomas’ rope squirrel, which we have now corrected (line 83).

Regarding the link between fire-footed rope squirrels and the mangabey outbreak, the co-detection of squirrel and MPXV DNA in mangabey faeces is just the final element that led us to conclude that very likely there was transmission of the virus from a squirrel to the mangabeys. The genomic data showing an identical virus in the mangabeys and in a fire-footed rope squirrel within a few weeks, and the behavioral observations of the exact same squirrel consumption are also supportive of this hypothesis.

23. Line 175: If the data exist, they should be included in this publication. Unpublished data are not acceptable evidence to support conclusions

We have removed the unpublished data and only left referenced information.

24. Line 183: personal communications are not acceptable evidence, please provide direct observation or cite from existing published literature to support this claim.

We have removed from the text the reference to “personal communication” (line251). This remains a direct field observation from the co-authors.

25. Lines 437 & 439: please include the primer/probe sequences in text so the work can be replicated without encountering a second paywall – particularly important for work relevant for and conducted in the Global South. The methods section lacks detail in general sufficient.

We have significantly enriched the methods section with details on primer sequences and PCR reactions, and have also added the squirrel mitogenome comparisons.

26. Supplemental table 4 is inadequate, please include basic taxonomic identification for each individual sampled; even if species-level identification is not possible the researchers should include highest resolution possible (e.g. genus or family) rather than a blank space.

We apologize for not providing this information in the first version of the table. We have added all available information in Supplementary table 4 as of June 2025, whether based on morphological identification (N=315) or molecular typing (N=364). We are still in the process of mitotyping all the rodents and shrews tested for orthopoxviruses, and plan to update this table with the most up-to-date mitotyping results at the time of a final revision.

Referee #4 (Remarks to the Author):

General comments

This study describes a remarkable investigation into the origin of a mpox outbreak in wild monkeys in West Africa. The diversity and quantity of laboratory methods directed towards the outbreak is impressive, going well beyond the norm for studies of wildlife diseases. Through longitudinal sampling, the authors stitch together a timeline and argue that the outbreak started when a monkey in the group ate an infected fire-footed rope squirrel. As the authors note, it is rare to have such temporal resolution in any wildlife disease outbreak investigation. The paper is well structured and well written, and the laboratory analyses are sophisticated and compelling. The figures are clear and interesting (especially the photos of animals in the forest), and colour is used well. The study will certainly rank among one of the most comprehensive wildlife disease outbreak investigations on record.

We thank the reviewer for the appreciations of our study.

27. A few overarching issues detract from the manuscript. First, in many sections the text reads as self-serving and self-congratulatory. The authors repeatedly stress that their foresight and superior research program in Ivory Coast is the reason why they were able to conduct this study, and that they are unusual in this respect. The effect is to be off-putting and to sound a bit “preachy” (e.g. lines 204-207). The data speak for themselves, so I suggest that the authors remove sections that tout their own preparedness and achievements. The value of long-term wildlife monitoring would make a good topic for a review paper or an opinion piece, but it is distracting here.

We thank the reviewer for their suggestion, which has prompted us to remove from the abstract and manuscript all sentences referring to the value of our long-term health wildlife health monitoring project.

28. More centrally, perhaps, is the conclusion that fire footed rope squirrels are a MPXV reservoir, which is the main finding of the study, as evidenced by the title. As interesting and compelling as the data are, they don't support this conclusion. In traditional terms, a reservoir is a population that naturally harbors a pathogen at high enough prevalence that it can serve as a source of infection for other species ("spillover"). Usually, the reservoir species carries the pathogen without clinical consequence, even though the pathogen can be deadly when transmitted to other species (e.g. Marburg virus in bats). In this study, the authors have documented virus infection in a single fire-footed rope squirrel that was found dead. The cause of death was unclear, but it is suspicious that MPXV might have killed it, in which case this goes against the idea that reservoirs typically aren't harmed by the pathogen for which they serve as a reservoir.

We thank the reviewer for raising this point, which has prompted us to enrich the text with a definition of reservoir (lines 78-80 of the revised manuscript). Following Haydon and colleagues (ref #8), we define a reservoir as a natural host in which a virus can circulate permanently and from which transmission to another host - humans in the case of zoonoses - is possible and documented. We agree that, in a number of instances, viruses show low pathogenicity in their natural hosts. Nonetheless, it is far from being a general rule, and many viruses are pathogenic in and sometimes exert enormous pressure on their natural hosts. A good example was another orthopoxvirus, the variola virus (the causative agent of smallpox), which induced high mortality in humans for centuries after our species became its natural host. Therefore even if the squirrel did die of MPXV infection it does not necessarily exclude the hypothesis of this species being a natural host, and following, a permanent reservoir for nonhuman primate and humans. Experimental studies of MPXV infection in rope squirrels have shown that the virus can replicate in these hosts and indeed be pathogenic and result in death, but a sizeable fraction of rope squirrel museum specimens was found to be infected with MPXV all the same – facts which suggest that rope squirrels are natural hosts of MPXV, which can be pathogenic. Further studies addressing MPXV infection dynamics in fire-footed rope squirrels are indeed required; however, we believe that even if MPXV had killed this squirrel, this would not be incompatible with squirrels being reservoirs and have now elaborated on our logic in the text.

29. Of course, there is the "multispecies reservoir concept" as well, in which a pathogen is maintained through transmission within and among several species (to which the authors allude in lines 187-189 but do not discuss further, as they perhaps should have). In that case, the fire footed rope squirrel may simply have been one of many reservoir species involved. Finally, the data are also consistent with the fire footed rope squirrel not being a reservoir at all. Perhaps there was an outbreak originating from another reservoir, and the fire footed rope squirrel 3 km south of the territory was a victim, just like the monkeys. Even if the monkey outbreak began with consumption of a fire-footed rope squirrel, maybe the monkey that ate the squirrel found it dead already, and the fire-footed rope squirrel had become infected from another, still unknown reservoir. To claim that the fire footed rope squirrel is a reservoir would require examining a large number of fire footed rope squirrels and a large number of other potential reservoirs in the forest. This study does not have such data, such that concluding that the fire footed rope squirrel is a reservoir is premature.

We agree with the reviewer that the sylvatic maintenance of MPXV probably relies on multiple hosts, and possibly also on their interactions. In the new version of the manuscript, we present all available data pointing towards distinct rope squirrel species and other African rodents as likely natural hosts of MPXV

(lines 83-92). In our opinion, besides the MPXV-infected fire-footed rope squirrel, the co-detection of MPXV and the same squirrel species DNA in the faeces of the mangabey sampled 3km away from the squirrel carcass (which is above the average home range of this squirrel species) shortly before a group-wide mpox outbreak, reinforces the hypothesis that MPXV was circulating in the local squirrel population in the fall of 2022. In the discussion of our findings, we now acknowledge that we cannot exclude that the MPXV-positive fire-footed rope squirrel may have acquired the infection from another host (lines 182-186). We do think that, as the reviewer suggests, further studies are needed to understand MPXV dynamics in this population and more broadly in rope squirrels and other small mammal species that may be involved in viral maintenance. That said, the notion that (fire-footed) rope squirrels would only be incidental intermediate hosts here is hard to reconcile with the high detection rate in museum specimens, which one would more parsimoniously explain by this virus permanently circulating in these hosts. Considering that the evidence available supports the notion that rope squirrels are natural hosts (lines 83-92), and our finding presented here that (1) a MPXV-infected fire-footed rope squirrel, (2) the exact same genomic variant of the virus in sympatric non-human primates, and (3) showing that mangabeys do eat squirrels combined with (4) finding MPXV and squirrel DNA in their feces prior to an outbreak, is in our view supportive of fire-footed rope squirrels having acted as reservoirs of MPXV for this specific outbreak.

30. Another point related to the above: how confidently can the 16s sequence obtained from diet analysis be attributed to the fire-footed rope squirrel? From lines 112-128 of the supplementary “diet analysis” section, it appears that the inference is based on a 91 bp segment of the 16s gene (which does BLAST closest to an unidentified squirrel from DRC). Could this segment be identical in multiple species of squirrels or even other rodents in the forest? As the authors’ BLAST analysis shows, online sequence databases are incomplete. It is therefore possible that this sequence is a perfect match to sequences of several local sciurids, including but not limited to the fire-footed rope squirrel.

We thank the reviewer for raising this point, which led us to look into 16S sequences of local sciurids as part of the manuscript revision. There are only 9 sciurid species known to exist in Côte d’Ivoire (IUCN Red List data), and the fire-footed rope squirrel is the only representative of the *Funisciurus* genus. We could retrieve 16S sequences from 5 of the 6 genera (a 16S sequence for *Epixurus* is not available, but this is not the most closely related genus to *Funisciurus*). In the alignment below, the reviewer will find sequences of the targeted 16S region from the fire-footed rope squirrel carcass, from the sooty mangabey fecal sample (named metabarcoding sequence), and from the most closely related sciurid sequences. As the reviewer can appreciate, the only 100% sequence match is with the TNP fire-footed rope squirrel. The next closest sequence was also generated from a fire-footed rope squirrel (of unknown provenance), and already differed at 10 positions, and other species sequences of course showed even greater divergence.

We conclude that this short fragment comprises enough information to unambiguously determine that the sequence found in the fecal sample is the result of the consumption of an individual squirrel belonging to the same species the carcass belonged to.

31. Finally, I'm not very familiar with the hybridization capture method described, but could it create biases in the MPXV genome sequences that make them seem identical when they are, in fact, not? Looking at how the method works, it appears that it cannot detect gene duplications or rearrangements and that it may preferentially enrich for conserved viral genome regions. If the identical nature of the MPXV genomes might be due to such biases, the authors should discuss this.

The hybridization capture method used is based on RNA probes that are designed on complete MPXV genomes. Since 2017, we have extensively used this technique to enrich for MPXV genomes (and many other viral and bacterial genomes) in samples of animal and human origin, and have detected genomic variation, including in nearly identical genomes that only differed in a 16bp deletion (Please see reference #18). We are therefore confident that the hybridization capture method did not bias the genomic information retrieved. We have now added more information on the genome coverage in the main text (please see lines 153-160 and 186-188) and a detailed description of the genome comparisons in the supplementary material (please see the supplementary information file). We also provide coverage plots as Extended Data Figure 5 and 6.

Directed comments

32. Lines 48-50. Consider deleting this sentence because it is a bit self-congratulatory and does not add information.

We removed the part of the sentence that pointed out to the value of long-term health monitoring.

33. Lines 52-53. The second phrase of this sentence is a bit too strongly worded, and the conclusion it makes strays too far from the data. Consider changing this phrase to a statement about risk, such as "exposure to these animals likely represents a risk for zoonotic transmission of mpox."

We have rephrased this sentence to welcome the suggestion of the reviewer. Please see lines 55-56.

34. Line 61. "Potent" may not be the best word choice here. Consider "useful," or simply delete.

We have removed the word "potent", as suggested.

35. Line 62. It's not clear to what "two pillars" refers. Change wording to clarify.

We have substituted the "two pillars" with "advances". We hope it will be clearer to the readers that we are referring to the upscaling of surveillance systems in endemic countries and to the discovery of APOBEC-3-induced signatures in the genomes of MPXVs sampled during long chains of human-to-human transmission. Please see line 65.

36 Line 68. The role of African rodents as MPXV reservoirs is more than an assumption. There is decent evidence to support this, beyond the two papers the authors cite. Rather than "formal proof to support this assumption is lacking," consider something more nuanced, such as "direct evidence from field studies has been rare."

We thank the reviewer for their suggestion, which picks up on the concerns expressed by reviewer #3. We acknowledge the lack of a broader context for some of our claims, which we now provide alongside a

revision of the wording. We have considerably re-written the first part of the manuscript, adding information on available evidence supporting the role of African rodents as natural hosts of MPXV. With “formal proof to support this assumption is lacking” we meant that, despite the accumulated evidence pointing at African rodents as being MPXV reservoirs (PCR results, serology etc), there has been no evidence clearly showing transmission of MPXV from an African rodent to another host. To clarify this notion, also following a suggestion of reviewer #3, we have enriched the introduction by reporting the findings of the 2003 USA mpox outbreak investigations. In this circumstance, despite identifying multiple species of MPXV-infected African rodents (dormice, giant pouched rats and rope squirrels), the investigations could not provide a definitive answer on which of the three African rodent species was the source of the MPXV that infected the prairie dogs and then humans. We believe the new version of the manuscript provides a broader overview of what we know thus far on MPXV circulation in potential natural hosts and reservoirs and hope to have addressed the reviewer’s concern with our textual clarifications. Please see lines 83-106 for details.

37. Lines 69-71. This sentence is too hyperbolic. Depending on what the reservoir(s) is/are, and what the sociopolitical context is, knowing the reservoir might not prevent MPXV transmission or thwart outbreaks. Consider re-writing in a more nuanced way, or delete.

Following the reviewer’s suggestion, we have toned down this statement by changing our statement to “knowing the reservoir host(s) may help managing spillover risk”. Please see lines 76-77.

38. Line 100. Which protocols were these, and is there a citation?

We apologize for not providing a reference for the mentioned protocols. In the revised version of the manuscript, we added a reference (#22) where we describe in detail the necropsy protocols that are in place at the Tai Chimpanzee Project and other sites where we conduct mortality surveillance on wildlife.

39. Line 108. Change “babies” to “infants,” to be consistent.

We have changed “babies” to “infants” in line 144.

40. Lines 187-189. Multispecies model. May be no single reservoir.

We agree with the reviewer, and have added significant information in the revised text to bring this scenario to the forefront. The multi-species model is exactly what we (and others) hypothesize based on the accumulated evidence. With this sentence, we highlight the importance of further investigating in this direction to have a better picture of which animal species are involved in the sylvatic maintenance of MPXV.

41. Figure 3. It looks like an adult animal (sixth row of the “Adults” section of the figure) tested positive for MPXV before the “index case” in red. This would seem to contradict the statement in lines 161-163. It’s difficult to try to figure this out from the supplemental tables.

We apologize for the misunderstanding of the presented data. The sample that appeared positive before the index case was just an artifact of many empty circles (i.e. negative samples) overlapping. We have modified Figure 3 to facilitate readers’ understanding of the displayed data. We hope the new version of the figure will clarify any doubts.

Dear Editor,

We sincerely thank you and the reviewers for the constructive feedback on the second version of our manuscript, and for giving us the opportunity to revise it further. We have carefully addressed the concerns raised during this additional round of review, which led us to modify the title of the manuscript and the main text to welcome the criticisms of Reviewer #4, and to perform phylogenetic analyses according to the methods suggested by the Reviewer #1.

Below, we provide a point-by-point response to the Reviewers' comments. For ease of reading, Reviewers' and Editor's comments are shown in blue, and our replies in black. We are submitting two versions of the manuscript: one with tracked changes and a clean version. The line numbers cited in our replies correspond to the tracked-changes version of the manuscript. To further aid clarity, we have numbered the modifications in the manuscript according to the corresponding comment number in this document.

We feel that the revisions have addressed all of the remaining concerns, and we hope you will find this version of the manuscript suitable for publication.

With our best regards,

Fabian H. Leendertz and Livia V. Patrono (on behalf of all coauthors)

Dear Professor Leendertz

Your manuscript, "Fire-footed rope squirrels (*Funisciurus pyrropus*) are a reservoir host of monkeypox virus (*Orthopoxvirus monkeypox*)", has now been seen by 3 referees. You will see from their comments below that while they find your work of interest, some important points are raised. We remain very interested in the possibility of publishing your study in Nature, but would like to consider your response to these concerns in the form of a revised manuscript before we make a final decision on publication.

As you can see, reviewer 1 has some outstanding concerns about the phylogenetic analysis.

During this further revision of our manuscript, we fully addressed the concerns of Reviewer 1 by conducting additional phylogenetic analyses, following their recommendations and using the suggested tools. These analyses, based on an expanded dataset and greater number of sites, confirmed our initial results. To reflect these updates, we revised the main text and the Materials and Methods sections, included new phylogenetic trees (both ML and MCC tree) generated from the expanded dataset, and added supplementary materials (Supplementary Table 7 and Supplementary Fig. 9) detailing the methodological approach used to obtain these results.

Referee 3 continues to feel that the conclusion that fire-footed rope squirrels are a reservoir host is not supported by these data - we would ask you to please reframe the paper according to the suggestions of the reviewer (it's still interesting, even without definitive statements about the reservoir).

We address below (point by point answer) the concerns raised by Referee 4 regarding fire-footed rope squirrels being a natural host of MPXV. We acknowledge that the data we present does not prove a permanent circulation of MPXV in fire-footed rope squirrels in the region, but rather presence of the virus in fire-footed rope squirrels in Tai National Park in a determined time window (end of 2022), and have revised our wording throughout the main text to reflect this. We have also rephrased the general conclusion drawn from additional studies on MPXV rodent hosts, now stating that based on all the evidence collected thus far, (rope) squirrels *may be* among the natural hosts of MPXV. However, we are convinced that our

data does show that fire-footed rope squirrels were the source of the outbreak in the mangabeys, and defend this hypothesis in detail in our point-to-point answer.

Overall, we have reframed all statements of fire-footed rope squirrels being a reservoir of MPXV, and propose a new title for the manuscript: “*Monkeypox virus (Orthopoxvirus monkeypox) outbreak in wild sooty mangabeys (Cercopithecus atys) following transmission from fire-footed rope squirrels (Funisciurus pyrropus)*”. Due to character limit restrictions in the file submission system, we had to shorten this title to “*Monkeypox virus outbreak in wild sooty mangabeys (Cercopithecus atys) following transmission from fire-footed rope squirrels (Funisciurus pyrropus)*”.

With these changes, we think we have addressed the criticisms of Reviewer #4. Shall the Editor feel that additional textual changes are required, we would be willing to further re-work the manuscript.

You will also need to make some editorial changes to your paper so that it is as brief as possible and complies with our Guide to Authors. We also strongly suggest that your revised manuscript has tracked changes, which is increasingly requested by referees to aid in their re-review.

The manuscript, figures, tables, and supplementary material have been revised to comply with the Guide to Authors. All recent changes have been tracked throughout the manuscript to facilitate re-review.

Point-by-point answer

Referee #1

1. We still have major concerns regarding the phylogenetic analyses that would benefit from further clarification and re-analyses.

Line 611: “Sites comprising ambiguities or gaps were removed in Geneious, resulting in an alignment of 134477 positions.” You have removed approximately 70 000 sites from the alignment. Were all sites containing ANY ambiguities or gaps in any sequence removed? The sites removed are substantially more than just the terminal repeats, which is the convention to mask (~7kb). What is the methodological justification for this? Missing information or gaps in columns will simply not contribute phylogenetic information for those sequences, while site patterns in other sequences add information and resolution. For terminal repeats and areas of low complexity, the correct approach is to mask your alignment. You can try <https://github.com/aineniamh/squirrel>. Again, the inference will likely hold, but there is no methodological justification for performing phylogenetics in this manner, especially as most of the inference is regarding divergence aka branch lengths.

The reviewer is right: in our initial analyses, we removed all sites containing ambiguities or gaps. We made the conservative assumption that these columns would be more likely to comprise sequencing/assembly errors. We considered that this arguably harsh choice was a reasonable trade off, since it still resulted in retaining >65% of the genomic alignment. To be on the absolute safe side, we performed the analyses requested by the reviewer on a larger alignment generated with squirrel. As the reviewer correctly predicted, our inference held. The reviewer will find the new results in lines (L) 189-198 and changes in the Material and Methods section (L632-656).

2. You cannot say sequences are identical, if you’re removing parts of the alignment without justification?

We want to clarify that the claim on sequence identity was never based on the trimmed alignment we used for phylogenetic analyses, but on direct pairwise alignments (L184-189).

3. All of the results, including the phylodynamic analyses, will be affected by this, particularly those addressing the “near-identicalness” of the genomes, and this issue should be directly addressed. I should also note that the consensus frequency threshold of 0.95 is quite conservative and may have excluded sites with lower read depth or true mixed populations, which are not necessarily indicative of contamination.

We agree with the reviewer that the thresholds we used to call unambiguous bases in our consensus are on the conservative end of the spectrum. That said, we believe that this is very much warranted in this case, and we note that there is no broadly accepted consensus on such thresholds in the literature, where different authors/groups implement different thresholds. Therefore, we decided to stick to our initial approach here. Of course, all raw reads will be publicly available and can be used by interested researchers to revisit our consensus and investigate potential within-host variation.

4. Line 626: Was the molecular clock unidentifiable or did it fail to converge? In either case, this is likely due to the exclusion of much of the phylogenetic backbone and therefore signal by focusing on just 23 sequences with low diversity. What was the justification for excluding other sequences from the analysis, particularly given that there was an overall positive temporal signal? I suspect that the rate heterogeneity in the reservoir host might not be strong enough in the shorter term in this subset to necessitate a relaxed clock, but rate heterogeneity should still be tested for. You can normally get an initial feel for it from the temporal regression – which should be included in the supplementary for evaluation as well. Additionally, the sampling time difference between the “near-identical” sequences should also be discussed in this context, to give the reader an expectation of divergence along the timescale given the evolutionary rate.

We repeated all analyses on the new, squirrel-generated alignment.

The dataset we used is still of reduced size but this reflects the sparse sampling of clade 2 II viruses. More genomes are available in the subclade of Nigerian sequences that also encompasses clade IIb and its progenitors. In our view, their inclusion would likely only increase rate variation (due to sequences exhibiting a heavy load of APOBEC3-induced mutations), without any obvious immediate benefit for the purpose of the analyses intended here.

This subclade is visible in Fig. 4 (based on 28 genomes in toto), but we excluded it from the molecular clock modeling after investigating temporal signal with PhyloStems. PhyloStems essentially runs a root-to-tip distance vs time regression for all possible subtrees in a given tree. This allows us to identify the particular subtree for which temporal signal is the strongest (using Rsq). In the new analysis, PhyloStems identifies the subclade of 24 sequences excluding the Nigerian sequences as being the subtree with the strongest temporal signal (Rsq: 0.71 vs 0.43 for the entire tree). We have added material to the Supplementary Information to support and illustrate our reasoning line: i. the regression on the entire tree obtained with TempEst, which notably allows us to pinpoint that 2 potential outliers precisely belong to the Nigerian subclade (Supplementary Figure 10), ii. the PhyloStems table identifying subtrees with Rsq>0.30 (Supplementary Table 7).

Using the new alignment, we were able to obtain reliable parameter estimates for the two molecular clock models (strict and uncorrelated lognormal relaxed clocks). In both cases, the heterochronous model outperformed the isochronous one, confirming the existence of a temporal signal. The relaxed clock model supported a higher mean rate (4.5×10^{-6} substitution per site per year) than the strict clock model (3×10^{-6} s.s-1.y-1) [which was itself slightly higher than our initial central estimate, although the latter did fall in the new 95% HPD]. This did not impact significantly our estimate of the time to the most recent common

ancestor of the sooty mangabey and squirrel-infecting viruses, which we dated again at 2021 (95% HPD: [2019-2022]). We added a sentence in the manuscript to highlight that under these rates and the assumption that they were epidemiologically related, the MPXV sequences derived from the squirrel and sooty mangabey were not expected to differ, and more generally rephrased the according paragraph to reflect both this reviewer and our considerations. The Reviewer will find the modified text in L189-204.

5. We believe the expanded discussion on alternative hosts, including other squirrel species, and the multi-species maintenance ecology sufficiently addresses our concerns in that area.

We thank the reviewers for recognizing our expanded discussion on alternative hosts and multi-species maintenance ecology, and we are pleased that it sufficiently addresses their concerns.

Referee #3:

6. The authors have sufficiently addressed my concerns and I congratulate them on this excellent and exciting work.

We thank the reviewer for their positive feedback and appreciation of our work.

Referee #4:

7. The authors have done an excellent job responding to critiques by providing new analyses and providing new explanatory text. The manuscript is extremely thorough and an impressive piece of work.

We sincerely thank the reviewer for their generous and encouraging comments. We are pleased that the additional analyses and revisions have improved the manuscript and that the work is viewed as thorough and valuable.

8. However, despite their rebuttal, the authors still are not justified in saying that they have shown the fire-footed rope squirrel to be a reservoir. Their responses to items 28 and 29 in the rebuttal only reiterate the remarkable story of sylvatic transmission in this outbreak; they do not strengthen the reservoir argument. There is little doubt that there is a connection between fire footed rope squirrels and this primate outbreak. However, that connection need not be explained by the fire footed rope squirrel being a reservoir. The fire footed rope squirrel might be a reservoir, but the data here (as impressive as they are) simply don't show that. The authors have added a definition of reservoir in lines 72-74, which is a good addition. That definition states that a criterion is "the virus can circulate permanently" in the host. The authors have not provided evidence of permanent (sustained) circulation of MPXV in the fire-footed rope squirrel. They have presented strong circumstantial evidence that a single fire footed rope squirrel nearby was infected with MPXV prior to the monkey outbreak.

Furthermore, the study of museum specimens that the authors cite is not about fire footed rope squirrels. It is about multiple species of rope squirrels, of which the fire footed is but one. In that study, 8/201 fire footed rope squirrels collected over 120 years across Central Africa tested positive for MPV. This shows a low period prevalence, and not "permanent circulation." As strong as the clues are that fire footed rope squirrels were involved in this outbreak, the conclusion that the fire footed rope squirrel is a reservoir is not supported by the data.

We thank the reviewer for their comments.

We understand the argument of the reviewer as being that: i. our data does not show fire-footed rope squirrels are a natural host in our system, ii. there is no solid evidence that it would be elsewhere.

Regarding the first point: what our data shows is that at least two fire-footed rope squirrels got infected with MPXV in the end of 2022. The second case derives from the co-detection of fire-footed rope squirrel and MPXV DNA in the first MPXV-positive fecal sample of the index sooty mangabey. There is no realistic alternative scenario that could plausibly explain such a co-detection by another mean than the infection of this fire-footed rope squirrel. We would like to stand by this interpretation, which we added in the last round of reviewing. We now stress that it shows the circulation of MPXV in this species *during this period* (L229-232). However, we acknowledge that identifying two infected individuals from a same species in such a short time window is insufficient to show permanent circulation in this species in the region. We now clearly state this in the first clause of the following sentence (L232-4).

Regarding the second point: we agree that the next question is to determine whether there is published data that support the notion of red-footed rope squirrels being natural MPXV hosts, and to which extent they do. The data available is limited, which entails some degree of subjectivity when interpreting it. For example, in the first review round, Referee 3 took the opposite stance to Reviewer #4's, and emphasized the need to give greater recognition to previous findings on potential MPXV natural hosts/reservoir species. Here, Reviewer #4 especially focuses on a specific museomic study that reported that 8 fire-footed rope squirrels were MPXV positive in a sample of 201 individuals. We would first argue that such a sample size for a single wildlife species is far from being negligible. We would then add that a 4% prevalence is, in fact, very high for a pathogen causing transient infections. As a point of comparison, it is estimated that 3-5% of the total population is infected with influenza viruses during the peak week of the influenza season in the northern hemisphere. In addition to this particular study and the others we discussed in the first paragraph we added in the introduction, a recent preprint reported detection of MPXV in two squirrels (including one rope squirrel) and one soft-furred mouse (<https://www.biorxiv.org/content/10.1101/2025.08.28.672325v1>). It is our opinion that the total evidence accumulated tips the interpretation further towards rope squirrels being likely natural MPXV hosts. To reflect these points and spell out more clearly what is our interpretation, we have now added the following: i. the number of detections in the museomic study, including in the two rope squirrel species mentioned in the text (L85), ii. a sentence reporting the most recent study on small mammals from the DRC (L86-89), and iii. a very cautiously phrased conclusion to the paragraph (L92-93): "*These repeated and independent findings suggest that (rope) squirrels may be among the natural hosts of MPXV.*"

Bringing together the first and second points, we think that we can write a more carefully worded conclusion to our results section (L234-7): "*Combined with the evidence of direct transmission reported in this study, the hypothesis that they serve as a reservoir of MPXV for wild nonhuman primates in TNP now appears very plausible.*"

9. Along these same lines, the original title of the manuscript is too strong. The authors should not say "are a reservoir." The proposed alternative title is also too strong. As impressive as the data are, they are circumstantial and do not demonstrate direct causality. The authors should change the title to something more objective, such as "Monkeypox virus outbreak in wild primates linked to transmission from fire-footed rope squirrels" (example only). It's clear that the authors want to state that they have identified the first MPXV reservoir (see lines74-75, "According to this definition, no MPXV reservoir has yet been identified"). However, as remarkable as the data are, the study still falls short of being able to attribute reservoir status to the fire-footed rope squirrel.

Please remove all statements in the title and elsewhere that go beyond the data presented in implicating the fire footed rope squirrel as a reservoir. The paper will be stronger for it.

We hope that we made clear that we agree with the reviewer that our data does not show that fire-footed rope squirrels are a natural host, and that the reviewer will appreciate that some remaining uncertainty is now clearly apparent in the manuscript. We agree that this should be embodied in a different title.

However, we respectfully disagree with the reviewer when they say that we did not show causation for this outbreak, and would in fact like the title to reflect clearly that we did. Indeed, the quantity and quality of the evidence that we present can only be said not to show cause under the harshest possible criteria. To recapitulate, here we show that: i local fire-footed rope squirrels were infected with MPXV shortly before the outbreak in sooty mangabeys, ii. mangabeys hunt and consume this species, iii. the first noninvasive sample positive for MPXV also contains fire-footed rope squirrel DNA, iv. the same individual is later positive for MPXV but not for fire-footed rope squirrel DNA, v. an outbreak of MPXV followed this index case infection. The only way to obtain better evidence is to cause an outbreak experimentally. We do believe that causation can be demonstrated beyond plausible doubt by longitudinal observational studies in the natural world (which is how it is routinely done in epidemiological studies), and we therefore propose that the title be: “*Monkeypox virus (Orthopoxvirus monkeypox) outbreak in wild sooty mangabeys (Cercocebus atys) following transmission from fire-footed rope squirrels (Funisciurus pyrropus)*”. Due to character limit restrictions in the file submission system, we had to shorten this title to “*Monkeypox virus outbreak in wild sooty mangabeys (Cercocebus atys) following transmission from fire-footed rope squirrels (Funisciurus pyrropus)*”.

In addition to this title change, we have revised the manuscript to avoid the overuse of the term “*reservoir*” which now only appears seven times. It is only used once in direct relation with our findings, in the aforementioned, much toned down conclusion (L234-7).